# Post-translational regulation of lipogenesis via AMPK-dependent phosphorylation of insulin-induced gene

Yamei Han[1], Zhimin Hu[1], Aoyuan Cui[1], Zhengshuai Liu[1], Fengguang Ma[1], Yaqian Xue[1], Yuxiao Liu[1], Feifei Zhang[1], Zehua Zhao[2], Yanyan Yu[3], Jing Gao[1], Chun Wei[4], Jingya Li[3], Jing Fang[1], Jia Li[3], Jian-Gao Fan[2], Bao-Liang Song [5] & Yu Li [1]

Insulin-induced gene (Insig) negatively regulates SREBP-mediated de novo fatty acid synthesis in the liver. However, the upstream regulation of Insig is incompletely understood. Here we report that AMPK interacts with and mediates phosphorylation of Insig. Thr222 phosphorylation following AMPK activation is required for protein stabilization of Insig-1, inhibition of cleavage and processing of SREBP-1, and lipogenic gene expression in response to metformin or A769662. AMPK-dependent phosphorylation ablates Insig's interaction with E3 ubiquitin ligase gp78 and represses its ubiquitination and degradation, whereas AMPK deficiency shows opposite effects. Interestingly, activation of AMPK by metformin causes an augmentation of Insig stability and reduction of lipogenic gene expression, and leads to the attenuation of hepatic steatosis in HFHS diet-fed mice. Moreover, hepatic overexpression of Insig-1 rescues hepatic steatosis in liver-specific AMPKα2 knockout mice fed with HFHS diet. These findings uncover a novel effector of AMPK. Targeting Insig may have the therapeutic potential for treating fatty liver disease and related disorders.

[1] CAS Key Laboratory of Nutrition, Metabolism and Food Safety, Shanghai Institute of Nutrition and Health, Shanghai Institutes for Biological Sciences, University of Chinese Academy of Sciences, Chinese Academy of Sciences, 200031 Shanghai, China. [2] Department of Gastroenterology, Xinhua Hospital, Shanghai Jiaotong University School of Medicine, 200092 Shanghai, China. [3] State Key Laboratory of Drug Research, Shanghai Institute of Materia Medica, University of Chinese Academy of Sciences, Chinese Academy of Sciences, 201203 Shanghai, China. [4] College of Biotechnology and Bioengineering, Zhejiang University of Technology, 310014 Hangzhou, China. [5] Hubei Key Laboratory of Cell Homeostasis, College of Life Sciences, Institute for Advanced Studies, Wuhan University, 430072 Wuhan, China. These authors contributed equally: Yamei Han, Zhimin Hu. Correspondence and requests for materials should be addressed to Y.L. (email: liyu@sibs.ac.cn)

Nonalcoholic fatty liver disease (NAFLD) develops when aberrant triglyceride accumulation in the liver is not compensated by the increased rate of fatty acid expenditure. Excessive hepatic de novo lipogenesis plays an important role in the development of NAFLD. Sterol-regulatory element-binding protein (SREBP) is a key transcription factor that regulates fatty acid synthesis[1]. SREBP is synthesized as precursor protein and retained in an inactive form in the endoplasmic reticulum (ER)[2], where it is bound to two other proteins, SREBP cleavage-activating protein (SCAP) and insulin-induced gene (Insig)[3,4]. When the cellular cholesterol levels are low, the SCAP–SREBP complex dissociates from Insig, then transports from ER to Golgi, where SREBP is cleaved by two membrane-bound proteases in a process called regulated intramembrane proteolysis (RIP). The released $NH_2$-terminal segment of SREBP translocates to the nucleus and stimulates lipogenic gene expression[5,6].

Insig is a potent inhibitor for the proteolytic process and maturation of SREBP via the retention of SCAP/SREBP complex in the ER[6]. Insig-1 is highly expressed in the liver, whereas Insig-2a is a liver-specific transcript of Insig-2[1,6]. Insig-1 and Insig-2 share similar function in that both isoforms cause ER retention of the SCAP/SREBP complex and exert a negative feedback control system on lipogenesis[7]. Transgenic overexpression of Insig-1 in the liver inhibits SREBP processing and lipogenesis[8]. In contrast, double knockout (DKO) of liver-specific Insig-1 and whole-body Insig-2 in mice (L-Insig-1, Insig-2−/−) results in increased lipogenic program and dramatic accumulation of lipid in the liver[9]. In sterol-depleted cells, Insig-1 protein is ubiquitinated and rapidly degraded by E3 ubiquitin ligase gp78 with a half-life of less than 30 min[10]. Interestingly, proteasomal degradation of Insig-1 is at least 15 times more rapid than Insig-2 due to the serine residues flanking the sites of ubiquitination[7]. However, the upstream signaling that mediates the post-translational regulation of Insig is poorly understood.

AMP-activated protein kinase (AMPK) monitors cellular energy status in response to nutritional variation in the environment[11]. Once activated, AMPK inhibits various anabolic pathways, stimulates catabolic pathways, suppresses ATP consumption, and increases ATP production to restore energy homeostasis[12,13]. We have previously identified that AMPK is a direct upstream kinase of SREBP. AMPK-dependent phosphorylation of SREBP-1c at ser372 site is sufficient and required for the inhibition of proteolytic cleavage and nuclear translocation of SREBP-1c[14]. However, SREBP-1c S372A mutation remains responsive to AMPK-mediated proteolytic cleavage and maturation of SREBP-1c, albeit the extent is less than wild-type (WT) SREBP-1c. These results suggest that additional AMPK substrates may directly or indirectly modulate SREBP-1c cleavage. Insig causes retention of the SCAP/SREBP complex in the ER, negatively regulates the cleavage of SREBP-1c, resulting in attenuation of lipogenic gene expression. However, whether AMPK regulates SREBP through Insig is not known.

We have recently identified transcriptional downregulation of Insig in the adaptive response to refeeding and under nutrient overload conditions through a novel metabolic cofactor CREBZF[15]. Here, we provide insights into the mechanism by which AMPK inhibits cleavage and activation of SREBP-1c via phosphorylation. Gain-of-function and loss-of-function studies characterize Insig as a critical effector in mediating AMPK and its agonist metformin in regulating lipogenesis and maintaining hepatic lipid metabolism. These in vivo and in vitro studies characterize that (1) AMPK is an upstream kinase of Insig; (2) AMPK-dependent phosphorylation of Insig ablates its interaction with E3 ubiquitin ligase gp78; (3) Thr222 phosphorylation of Insig-1 is essential for AMPK to enhance Insig-1 activity and inhibit SREBP-1c proteolytic cleavage and target lipogenic gene expression; (4) Metformin attenuates hepatic steatosis in part through activation of Insig.

## Results

**Metformin stimulates AMPK and Insig activity in mouse livers.** Several clinical studies indicate that the antidiabetic drug metformin might be of benefit for the treatment of NAFLD in humans[16–19]. To investigate the efficacy and mechanisms of metformin on hepatic steatosis and lipid metabolism, 8-week-old male C57BL/6 mice were placed for 8 weeks HFHS diet, and followed by a dose of metformin (50 mg/kg/day), which has been shown to lower hepatic steatosis in high-fat diet-fed mice[20]. As shown in Fig. 1a and b, administration of metformin caused a significant reduction of excess fat accumulation in hepatic intracellular vacuoles, as determined by hematoxylin and eosin (H&E) and oil red O staining. Consistently, liver triglyceride and cholesterol levels were reduced by administration of metformin compared with the control mice. These results suggest that metformin is sufficient to improve hepatic steatosis in diet-induced obese (DIO) mice.

To investigate whether AMPK might be responsible for the protective effects of metformin, hepatic AMPK activity was assessed by determining the phosphorylation levels of AMPK and its downstream substrate acetyl-CoA carboxylase (ACC). As shown in Fig. 1c and d, hepatic phosphorylation of AMPK and ACC were increased significantly by metformin, which were further confirmed by immunohistochemical analysis of liver sections. Interestingly, as shown in Fig. 1e and f, protein levels of both Insig-1 and Insig-2 were markedly increased up to around two-fold by metformin administration as evidenced by immunoblotting analysis and immunohistochemical staining, which was inversely correlated with a reduced expression of nuclear active form of SREBP-1 and fatty acid synthase (FAS). Notably, the expression of nuclear SREBP-2 was also reduced by metformin (Supplementary Fig. 1C). These results suggest a correlation between metformin-mediated activation of AMPK and Insig, which may play a role in the regulation of hepatic lipid metabolism and improvement of hepatic steatosis.

**Protein levels of Insig-1 and Insig-2 are increased by AMPK.** To test whether AMPK regulates Insig in vitro, metformin was used to treat HEK293 cells transfected with plasmid encoding Insig. Real-time PCR analysis was performed to determine the effects of metformin on transcription levels of Insig-1. Surprisingly, mRNA levels of Insig-1 were not obviously changed by metformin (Supplementary Fig. 1A). Interestingly, protein levels of Insig-1 were markedly increased by metformin, proteasome inhibitor MG132, or AICAR in HEK293 cells (Fig. 2a, b). In addition, treatment with an adenovirus encoding a myc-tagged constitutively active form of AMPK (Ad-CA-AMPK) in HepG2 hepatocytes had the similar effect (Fig. 2c). These results indicate that AMPK may regulate the expression of Insig-1 through post-translational modification in vitro.

Furthermore, to explore whether AMPK also regulates Insig-2 protein levels, AMPK agonists metformin and AICAR, and adenovirus encoding Ad-CA-AMPK were used to treat cells transfected with Insig-2 plasmid. Consistent with Insig-1, protein levels of Insig-2 were increased by AMPK, whereas Insig-2 mRNA levels were not obviously changed (Figs. 2d–f and S1B). Taken together, these data indicate that AMPK is sufficient to stimulate both Insig-1 and Insig-2 activity in vivo and in vitro likely through a post-translational regulation.

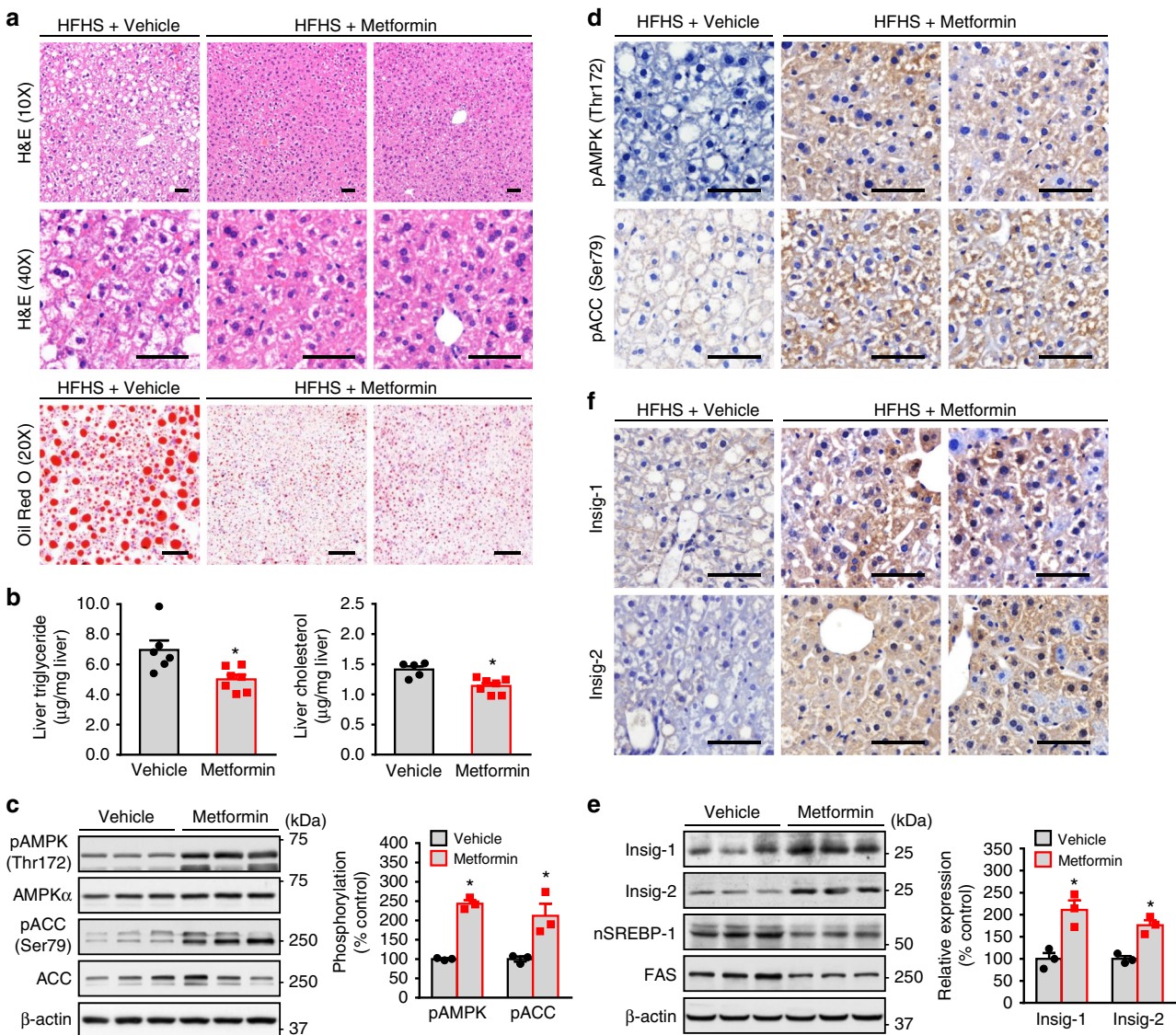

**Fig. 1** Administration of metformin stimulates AMPK activity and increases Insig levels to attenuate hepatic steatosis in high-fat, high-sucrose diet-fed mice. Eight-week-old male C57BL/6 mice were fed with high-fat, high sucrose (HFHS) diet for 8 weeks, and then fed with HFHS diet supplemented with daily metformin (50 mg/kg/day) or vehicle (PBS) treatment via intraperitoneal injection for 8 weeks. **a, b** Metformin attenuates hepatic steatosis in HFHS diet-fed mice. **a** Representative H&E and oil red O staining of liver sections are shown (scale bar, 50 μm). **b** Liver triglyceride and cholesterol levels were assessed. **c, d** AMPK activity is increased by metformin in the liver. **c** Immunoblots were performed, and the band intensity was quantified by densitometry. **d** Representative immunohistochemical staining of the liver sections with pAMPK and pACC antibodies. **e, f** The protein levels of Insig-1 and Insig-2 are increased in the liver of mice treated with metformin. **e** Immunoblots were performed, and the band intensity was quantified by densitometry. **f** Representative immunohistochemical staining of the liver sections with Insig-1 and Insig-2 antibodies. Scale bars, 50 μm. Data are presented as the mean ± SEM, unpaired two-tailed Student's $t$-test, $n = 4$–6, *$p < 0.05$, vs. vehicle

**AMPK inhibits the degradation of Insig protein**. Given that regulated degradation of Insig is central for SREBP processing and cellular lipid homeostasis[10], we hypothesize that AMPK may regulate Insig via the ubiquitin-proteasome pathway (UPP) of protein degradation. To test this hypothesis, HEK293 cells were transfected with myc-tagged Insig-1 or FLAG-tagged Insig-2 together with or without HA-tagged ubiquitin (HA-Ub). As shown in Fig. 3a and b, the ubiquitination of Insig-1 and Insig-2 were observed in the presence of HA-Ub. Strikingly, AMPK activation by A769662 (6,7-Dihydro-4-hydroxy-3-(2′-hydroxy[1,1′-biphenyl]-4-yl)-6-oxo-thieno[2,3-b]pyridine-5-carbonitrile)[21] significantly reduced ubiquitination levels of both Insig isoforms. These data indicate that AMPK activation decreases ubiquitination levels of Insig.

We next investigated whether AMPK regulates Insig degradation. Given that Insig-1 is degraded much more rapid than Insig-2[10], we focused on the study of AMPK's effects on Insig-1 protein stability. HEK293 cells stably expressing Insig-1 were treated with cycloheximide, a eukaryote protein synthesis inhibitor, to measure the half-life of Insig-1. As shown in Fig. 3c and d, compared with cells treated with vehicle, the proteasome-mediated degradation of Insig-1 in the presence of metformin, AICAR, or A769662 were significantly reduced, which resulted in an induction of half-life from 0.5 to 0.9 h, 1.4, or 1.7 h, respectively. Together, these data suggest that AMPK increases the stability of Insig protein via reducing UPP-mediated degradation.

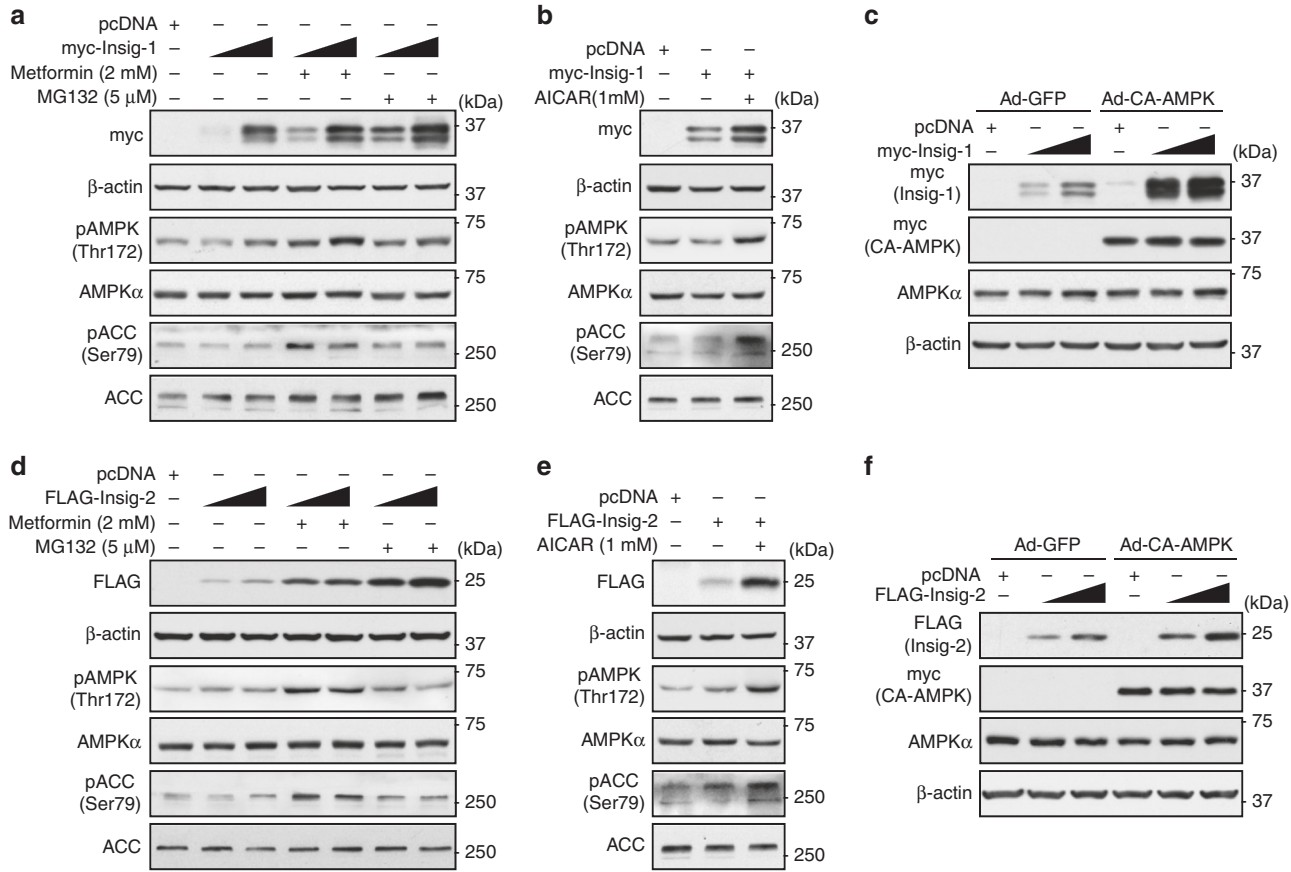

**Fig. 2** Activation of AMPK enhances the protein levels of both Insig-1 and Insig-2. **a–c** AMPK agonists increase the protein levels of Insig-1. HEK293 cells were transfected with pcDNA or myc-tagged Insig-1 for 24 h, followed by treatment with 2 mM metformin or 5 μM MG132 (**a**), or 1 mM AICAR (**b**) as indicated; and HepG2 cells were transfected with pcDNA or myc-tagged Insig-1 for 24 h, and then infected with adenoviruses encoding constitutive active AMPK (CA-AMPK) or GFP for 48 h (**c**). Immunoblots were performed. **d–f** AMPK agonists increase the protein levels of Insig-2. HEK293 cells were transfected with pcDNA or FLAG-tagged Insig-2 for 24 h, followed by treatment with 2 mM metformin or 5 μM MG132 (**d**), 1 mM AICAR (**e**); and HepG2 cells were transfected with pcDNA or FLAG-tagged Insig-2 for 24 h, and then infected with adenoviruses encoding CA-AMPK or GFP for 48 h (**f**). Immunoblots were performed

**AMPK inhibits the interaction between Insig and gp78.** It has been reported that protein levels of both Insig-1 and Insig-2 are dramatically elevated in the liver of liver-specific gp78 knockout (gp78 LKO) mice fed with chow diet[22], in which gp78 may also increase the degradation of both Insig-1 and Insig-2 through ubiquitination pathway. However, gp78 was characterized as an E3 ligase for Insig-1, instead of Insig-2, in Chinese hamster ovary (CHO) cells and human fibroblasts[7,23]. To investigate whether gp78 regulates expression levels of both Insig isoforms, HEK293 cells were transfected with Insig-1 and Insig-2 together without or with gp78. As shown in Fig. 4a and b, treatment with gp78 caused a profound reduction of Insig. Interestingly, AMPK activation by metformin partially reversed the inhibitory effects of gp78 on both Insig-1 and Insig-2.

To investigate whether AMPK-stimulated Insig expression is mediated via the regulation of expression levels of the ligase gp78, real-time PCR analysis was performed. As shown in Supplementary Fig. 2, mRNA levels of gp78 were not changed in cells by metformin or other AMPK agonists. Next, to test the hypothesis that AMPK might upregulate Insig activity through regulating the protein interaction between Insig and gp78, co-immunoprecipitation (Co-IP) analysis was performed. As shown in Fig. 4c and d, treatment with A769662 indeed ablated the association between Insig-1 or Insig-2 with gp78. Notably, phosphorylation of AMPK and induction of Insig expression were observed in cell lysates. Taken together, these data

demonstrate that gp78 may act as the E3 ligase for both Insig isoforms, and AMPK is sufficient to attenuate gp78-mediated suppression of Insig.

**AMPK associates with and phosphorylates Insig-1.** We hypothesized that AMPK might increase Insig activity though phosphorylation. To define Insig as a novel target of AMPK, Co-IP analysis was performed. As shown in Fig. 5a and b, Insig-1 was co-expressed with GST-tagged AMPKα1 in a dose-dependent manner. Interestingly, the interaction between AMPKα1 and Insig-1 was greatly enhanced when AMPK was activated by A769662. As shown in Fig. 5c, phosphorylation of Insig-1 as measured by antibody against phosphoserine/threonine (pSer/Thr) was observed in the basal condition. Strikingly, AMPK activation by A769662 caused a profound induction of Insig-1 phosphorylation, whereas AMPK inhibition by compound C (6-[4-(2-Piperidin-1-yl-ethoxy)-phenyl])-3-pyridin-4-yl-pyyrazolo [1,5-a] pyrimidine)[24] had the opposite effect. Notably, phosphorylation levels of AMPK in response to A769662 and compound C were evidenced. These results demonstrate that Insig-1 is regulated by AMPK through phosphorylation.

We further investigated the AMPK phosphorylation sites in Insig-1. Liquid chromatography–tandem mass spectrometry (LC–MS/MS) analysis was performed and a few putative phosphorylation sites in human Insig-1 protein were revealed,

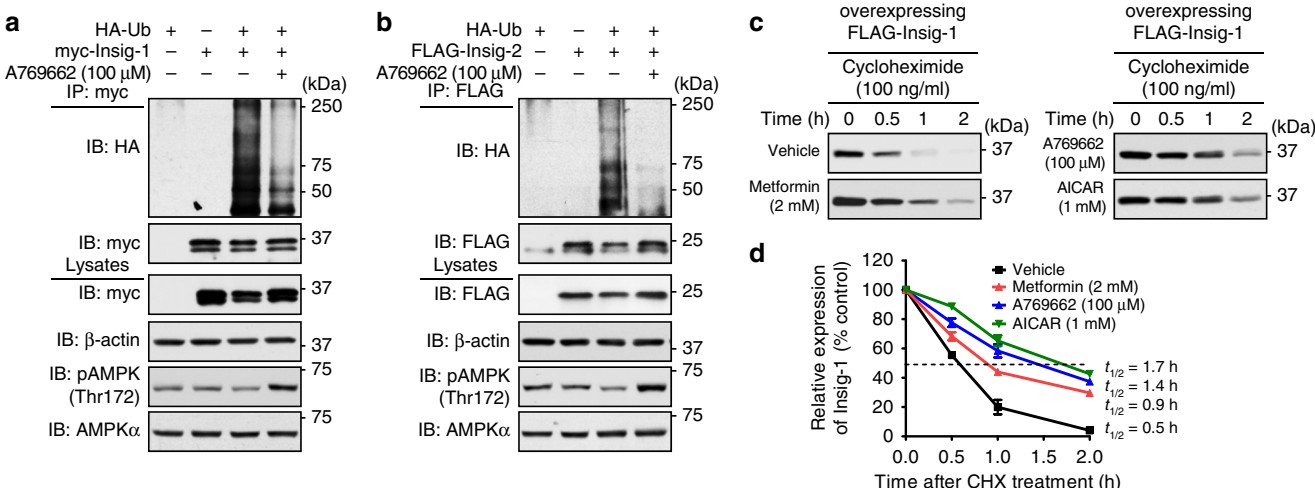

**Fig. 3** Administration of AMPK agonist represses the ubiquitination and degradation of Insig. **a, b** AMPK agonist decreases the ubiquitination levels of Insig-1 and Insig-2. Plasmids encoding myc-tagged Insig-1 (**a**) or FLAG-tagged Insig-2 (**b**) were co-transfected with HA-tagged ubiquitin (HA-Ub) for 24 h, followed by treatment with 100 μM A769662 for 4 h. The cell lysates were incubated with myc or FLAG antibodies and purified with protein A/G-Sepharose beads. The precipitates and lysates were individually immunoblotted with indicated antibodies. **c, d** AMPK agonists reduce the degradation rate of Insig-1. HEK293 cells stably expressing FLAG-tagged Insig-1 were treated with 100 μM A769662, 2 mM metformin, and 1 mM AICAR, followed by incubation in the medium containing 100 ng/ml cycloheximide for the indicated time. Data are presented as the mean ± SEM, $n = 4$

including Ser43, Ser46, Ser74, Thr216, Thr222, and Ser276. Strikingly, the mutation of T222A-eliminated metformin-caused induction of Insig-1 (Figs. 5e, S3A and S3B), suggesting that Thr222 site is essential for AMPK's stimulating effects on Insig-1. Importantly, the phosphorylation of Insig-1 by AMPK was ablated by the nonphosphorylatable T222A mutant (Fig. 5f), indicating that Thr222 is specialized phosphorylated by AMPK. Notably, the mutation of other putative AMPK phosphorylation sites predicted via sequence alignment, including Ser125, Ser178, Ser190, Ser209, Ser238 to nonphosphorylatable alanine did not change AMPK's effects on the stabilization of Insig-1. As shown in Supplementary Fig. 3C and Fig. 5g, compared with cells treated with vehicle, the proteasome-mediated degradation of WT Insig-1 and S74A mutant in the presence of A769662 were significantly reduced, whereas T222A mutant was resistant to A769662-caused protein stabilization. These data indicate that AMPK phosphorylation at Thr222 site is essential for the stabilization of Insig-1 protein. Together, these results suggest that AMPK may associate with and phosphorylate Insig-1 to inhibit the interaction between Insig-1 and its E3 ligase gp78, and lead to an augmentation of Insig-1 activity and reduction of fatty acid synthesis.

**AMPK suppresses the cleavage of SREBP-1c S372A mutant.** Next, we sought to study the mechanisms of AMPK on the proteolytic processing of SREBP-1c. Given that the positive feedback regulation of SREBP through the SRE element in its own promoter[25], it is technically difficult to determine whether the change of nuclear SREBP fragment is due to the regulation of proteolytic cleavage or transcription. We therefore sought to generate exogenous FLAG-tagged SREBP expression system, in which FLAG-tagged SREBP-1c is driven by a cytomegalovirus (CMV) promoter, which is not affected by autoregulation of SREBP. As shown in Fig. 6a, this cleavage assay showed that AMPK activation by metformin or A769662 were sufficient to inhibit the mature, active nuclear form of SREBP-1c, whereas AMPK inhibition by compound C resulted in the opposite effect, which is consistent with AMPK-increased stabilization of Insig in HEK293 cells. Moreover, the cellular fractionation was performed. As shown in Fig. 6b, consistent with the measurement

using whole lysates, the proteolytic processing and nuclear fragment of exogenous SREBP-1c were strikingly decreased by treatment with metformin, whereas the expression of precursor SREBP-1c was not obviously changed. These results suggest that the SREBP stable cell system is a unique and valuable tool for the study of SREBP-1 proteolytic cleavage.

It has been reported that AMPK phosphorylates SREBP-1c at Ser372 site to repress SREBP-1 proteolytic cleavage and attenuate hepatic lipogenesis[14]. To investigate the relative contribution of Ser372 to AMPK-caused inhibition of SREBP-1c cleavage, HepG2 cells stably expressing nonphosphorylatable SREBP-1c S372A mutant were generated. As shown in Fig. 6c and d, metformin treatment caused a potent reduction of SREBP-1c cleavage in WT SREBP-1c stable cells. Strikingly, SREBP-1c S372A mutant remained responsive to metformin-inhibited proteolytic cleavage significantly, although the magnitude of the reduction is relatively lower compared with that in WT stable cells. Notably, phosphorylation of AMPK and ACC were observed. These results indicate that AMPK may inhibit SREBP-1c cleavage via a novel mechanism. Moreover, the efficacy of metformin on SREBP-1c S372 mutant cleavage was compared with that of adenoviruses expressing a low-dose of Insig-1. As shown in Supplementary Fig. 4, treatment with 2 mM metformin was sufficient to repress SREBP-1c S372 cleavage. Interestingly, treatment of Ad-Insig-1 that mimics metformin-stimulated endogenous Insig-1 expression resulted in a similar reduction of SREBP-1c S372 cleavage. These results suggest that AMPK activation by metformin may inhibit SREBP-1c S372A cleavage via stabilization of Insig in hepatocytes. As shown in Fig. 6e, the amount of nuclear form of SREBP-1c derived from HepG2 cells stably expressing FLAG-tagged SREBP-1c S372A was largely reduced by WT Insig-1 when treated with A769662, accompanied by an reduction of the lipogenic enzyme FAS. Conversely, the T222A mutant diminished the inhibitory effects of A769662 on SREBP-1c S372A cleavage and expression of FAS. Together, these data suggest that AMPK, via Thr222 phosphorylation, inhibits proteasome-mediated degradation of Insig-1, and leads to a reduction of SREBP-1c cleavage processing and lipogenic gene expression.

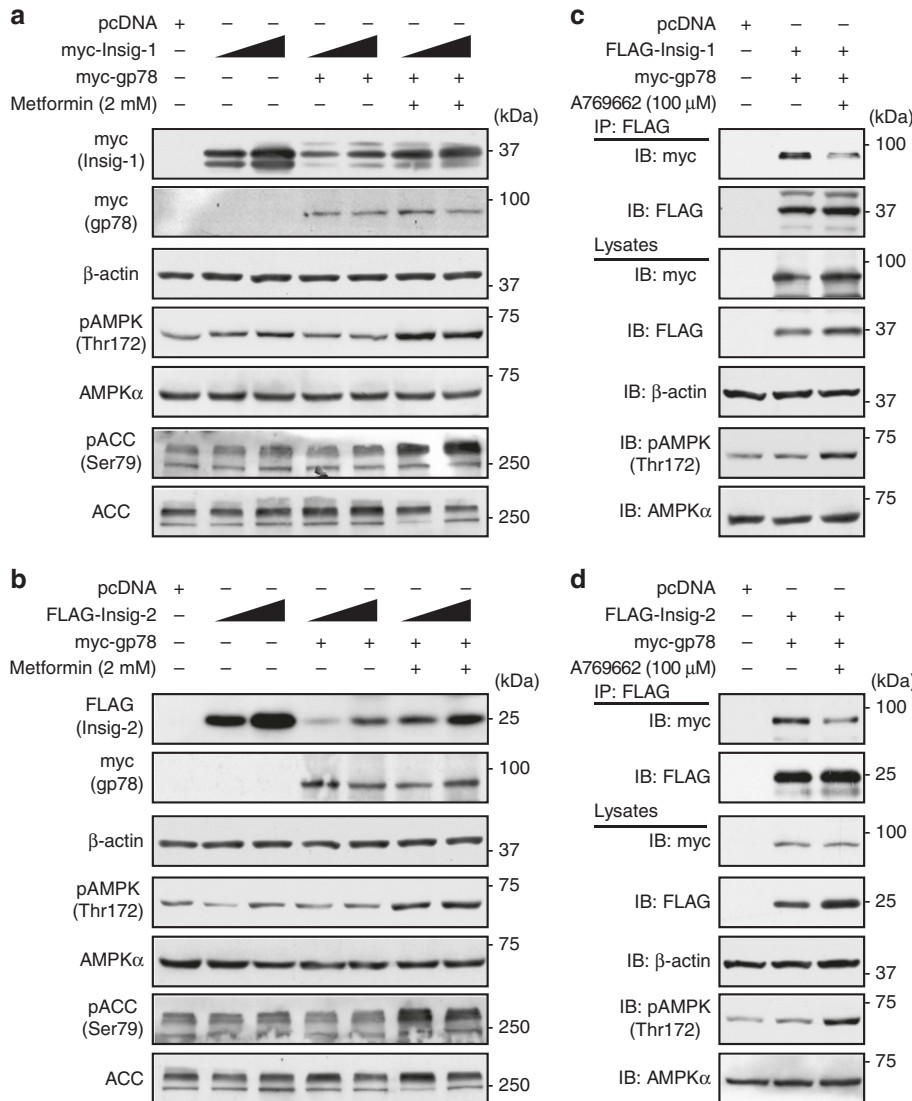

**Fig. 4** Activation of AMPK ablates the interaction between Insig and the ubiquitin ligase gp78. **a**, **b** Metformin reverses the gp78-mediated degradation of Insig-1 (**a**) and Insig-2 (**b**). myc-Insig-1 or FLAG-Insig-2 was co-transfected with myc-gp78 in HEK293 cells for 24 h, followed by treatment with 2 mM metformin for 24 h. Immunoblots were performed. **c**, **d** AMPK-specific agonist A769662 inhibits the interaction between Insig and gp78. FLAG-tagged Insig-1 (**c**) or Insig-2 (**d**) was co-transfected with myc-tagged gp78 in HEK293 cells for 24 h, followed by treatment with 100 μM A769662 for 8 h. The cell lysates were incubated with FLAG antibodies and purified with protein A/G-Sepharose beads. The precipitates and lysates were individually immunoblotted with indicated antibodies

To further test the functional consequence of AMPK-SREBP-1c S372 mutant on hepatic lipid metabolism, oil red O staining was performed. As shown in Fig. 6f and g, compared with control cells expressing empty vector (EV), excessive accumulation of lipid was observed in HepG2 cells expressing WT SREBP-1c. Treatment with metformin caused a marked reduction. Importantly, consistent with the SREBP-1c cleavage assay in Fig. 6c and d, SREBP-1c S372A mutant stably expressed in human HepG2 cells remains responsive to AMPK agonist metformin-caused reduction of lipid accumulation. Taken together, these studies suggest that AMPK phosphorylates and increases Insig activity to inhibit SREBP cleavage and attenuate hepatic steatosis in hepatocytes.

**Hepatic steatosis of AMPKα2 LKO mice is reversed by Insig-1**. To determine the causal link between AMPK and Insig, AMPK loss-of-function approaches were performed. As shown in Fig. 7a,

activation of AMPK by metformin, A769662 and AICAR enhanced protein levels of Insig-1 in AMPK+/+ hepatocytes, whereas these effects were abrogated in AMPKα1α2 double knockout hepatocytes (Fig. 7a). Notably, AMPK deficiency was confirmed by the phosphorylation of AMPK and ACC. Consistently, as shown in Fig. 7b, AMPK activation by A769662 increased protein levels of Insig-1 in SRD-13A cells, a SCAP-deficient CHO cell line[26]. In contrast, AMPK inhibition by compound C caused a profound reduction of Insig-1 protein levels. To further investigate whether AMPK inhibition causes an increased degradation of Insig, cycloheximide assays were performed in HEK293 cell stably expressing Insig-1. AMPK inhibition by compound C significantly promoted the degradation of Insig-1, resulting in the half-life change of Insig-1 from 30 min in vehicle-treated cells to around 20 min in compound C-treated cells (Fig. 7c and d).

Given that inhibition of AMPK accelerates Insig-1 degradation and reduces its protein levels, rescue assays with adenovirus

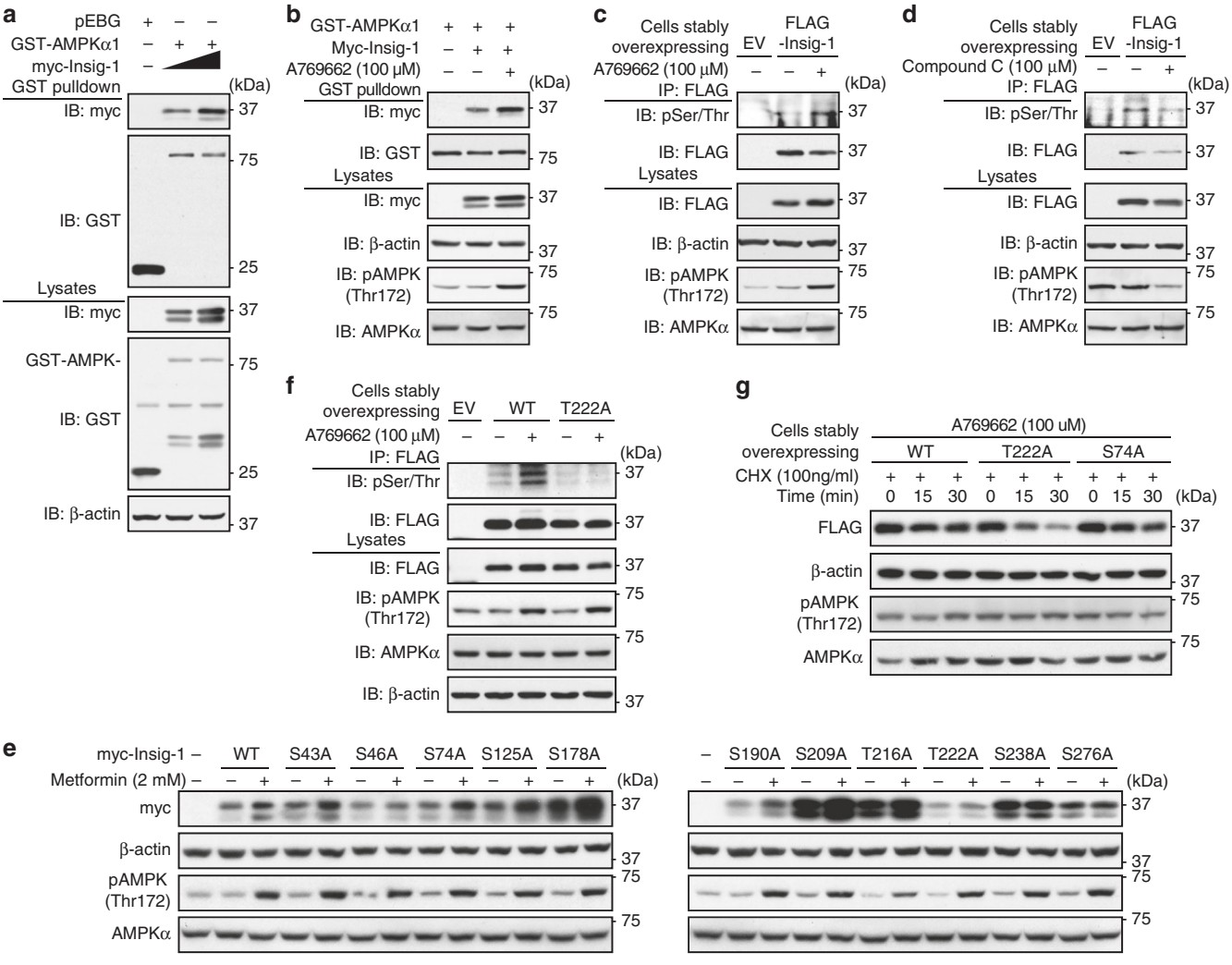

**Fig. 5** Insig-1 is a downstream target of AMPK. **a** AMPKα1 subunit associates with Insig-1 in a dose-dependent manner. HEK293 cells were transfected with myc-Insig-1, GST-AMPKα1, and pEBG (empty vector) as indicated for 48 h, and purified with GSH Sepharose beads. **b** AMPK activation by A769662 enhances the association between AMPKα1 and Insig-1. HEK293 cells were transfected with myc-Insig-1 and GST-AMPKα1 as indicated for 24 h, followed by treatment with 100 μM A769662 for 8 h, and then purified with GSH Sepharose beads. The precipitates and lysates were immunoblotted with antibodies against myc or GST. **c**, **d** Insig-1 is phosphorylated by AMPK. Insig-1 is phosphorylated in response to A769662 (**c**), whereas the phosphorylation level is inhibited by compound C (**d**). HEK293 cells stably expressing FLAG-tagged Insig-1 or empty vector (EV) were treated with 100 μM A769662 or 20 μM compound C for 30 min. The cell lysates were incubated with FLAG antibodies and purified with protein A/G-Sepharose beads. The precipitates were immunoblotted with antibodies against phosphoserine/threonine (anti-pSer/Thr). **e** The effects of metformin on various Insig-1 mutants. HEK293 cells were transfected with pcDNA or myc-tagged Insig-1 nonphosphorylatable mutant plasmids for 24 h, followed by treatment with 2 mM metformin for 24 h. Immunoblots were performed. **f** AMPK phosphorylates human Insig-1 at Thr222. HEK293 cells stably expressing FLAG-tagged Insig-1 or T222A mutant were treated with 100 μM A769662 for 30 min. The cell lysates were incubated with FLAG antibodies and purified with protein A/G-Sepharose beads. The precipitates were immunoblotted with antibodies against anti-pSer/Thr. **g** T222A mutant, but not WT or S74A mutant of Insig-1, is unstable and resistant to AMPK activation. HEK293 cells stably expressing FLAG-tagged WT, T222A, or S74A mutant of Insig-1 were treated with 100 μM A769662, followed by incubation in the medium containing 100 ng/ml cycloheximide for the indicated time

encoding Insig-1 (Ad-Insig-1) were performed in liver-specific AMPKα2 knockout (AMPKα2 LKO) mice. As shown in Fig. 7e, f, compared with the WT littermates (albumin-Cre negative, AMPKα2^flox/flox), hepatic AMPKα2 deficiency caused mild steatotic phenotypes in mice fed with HFHS diet as evidence by H&E staining, oil red O staining and triglyceride levels. Notably, the knockout efficiency of AMPKα2 specifically in the liver was verified by immunoblots. Strikingly, hepatic overexpression of Insig-1 rescued hepatic steatosis in AMPKα2 LKO mice. Next, we further determined whether reduction of SREBP-1c signaling and lipogenic program is responsible for improved hepatic steatosis in Ad-Insig-1-treated mice. The expression of Insig-1 in the liver was evidenced by RT-PCR (Fig. 7g). mRNA levels of SREBP-1's

lipogenic targets and enzymes involved in fatty acid and triglyceride synthesis, including SREBP-1c, ATP citrate lyase (ACLY), acetyl-CoA carboxylase 1 (ACC1), FAS, stearoyl CoA desaturase 1 (SCD1), glycerol-3-phosphate acyltransferase (GPAT), diglyceride acyltransferase 2 (DGAT2) were significantly decreased by treatment with Ad-Insig-1. Taken together, these results strongly suggest an essential role of Insig in mediating AMPK's beneficial effects on lowering hepatic steatosis and maintaining lipid metabolism in the liver.

## Discussion
This report demonstrates that Insig may act as a novel substrate for AMPK in regulating hepatic lipogenesis and maintaining lipid

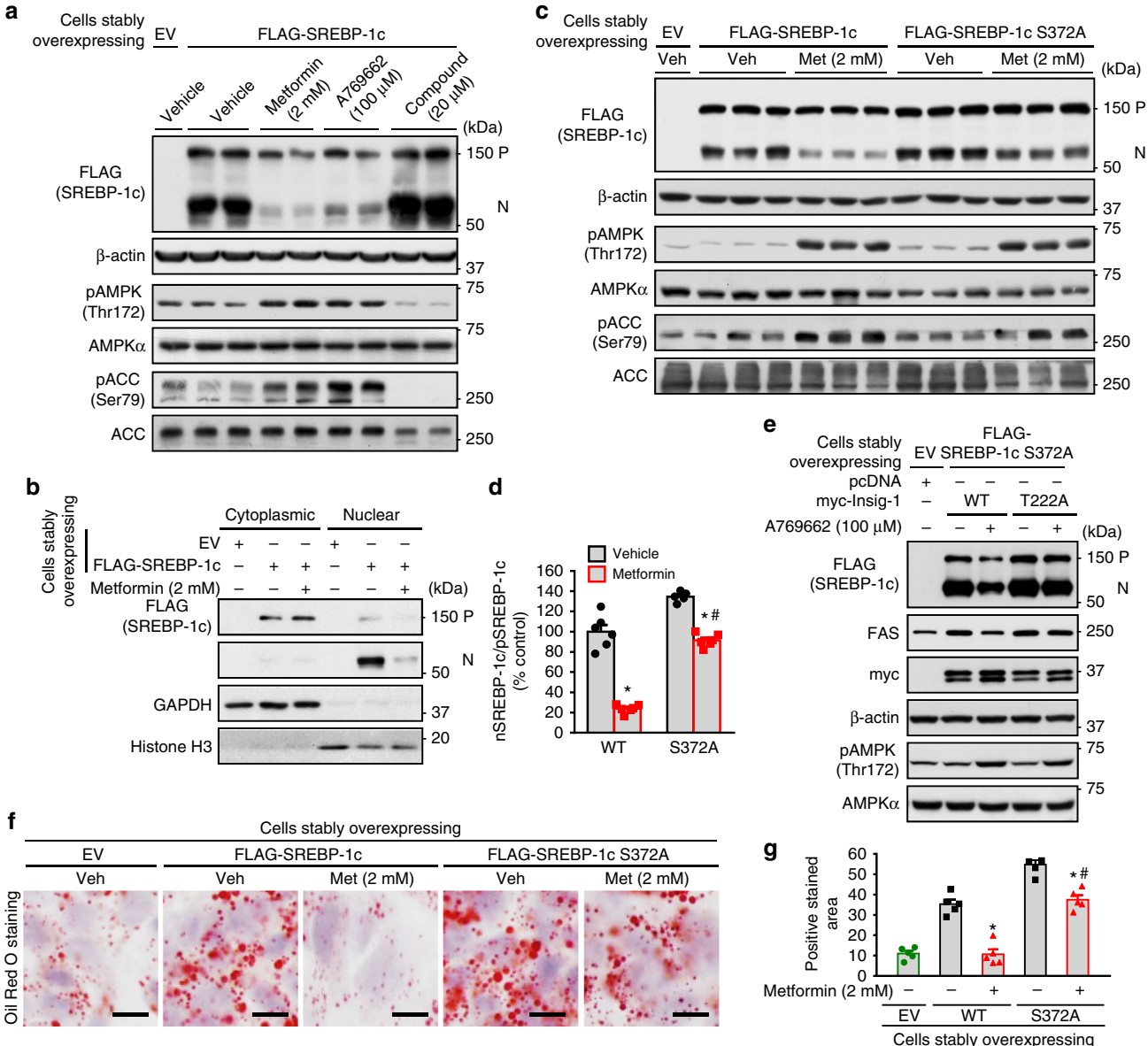

**Fig. 6** SREBP-1c S372A mutant stably expressed in human HepG2 cells remains responsive to AMPK-inhibited proteolytic cleavage. **a** The cleavage processing of exogenously wild-type (WT) SREBP-1c stably expressed in human HepG2 hepatocytes is sensitive to the treatment of AMPK agonists or inhibitor. HepG2 cells stably expressing FLAG-tagged SREBP-1c were treated with 2 mM metformin for 24 h, 100 μM A769662 for 8 h, or 20 μM compound C for 24 h. Immunoblots were performed, and the precursor and active nuclear form of SREBP-1c was visualized using FLAG antibody. **b** The nuclear translocation of SREBP-1c is prevented by metformin. HepG2 cells stably expressing FLAG-tagged WT SREBP-1c were treated with 2 mM metformin for 24 h. Cellular fractionation was performed, followed by immunoblots to evaluate the levels of SREBP-1c in the cytoplasm and nucleus. **c**, **d** SREBP-1c S372A mutant stably expressed in human HepG2 cells remains responsive to AMPK agonist metformin-inhibited proteolytic cleavage. HepG2 cells stably expressing FLAG-tagged WT SREBP-1c or SREBP-1c S372 mutant were treated with 2 mM metformin for 24 h. Immunoblots were performed. **e** Thr222 phosphorylation of Insig-1 is required for the inhibition of SREBP-1c cleavage and lipogenic gene expression by A769662. HepG2 cells stably expressing FLAG-tagged SREBP-1c S372 mutant were transfected with pcDNA or myc-tagged WT Insig-1 or T222A mutant for 24 h, and then treated with 100 μM A769662 for 8 h. Immunoblots were performed. **f**, **g** SREBP-1c S372A mutant stably expressed in human HepG2 cells remains responsive to AMPK agonist metformin-inhibited lipid accumulation. Cells were treated with 2 mM metformin for 24 h. Representative oil red O staining (**f**) and the quantification of are shown (**g**). Scale bars, 10 μm. Data are presented as the mean ± SEM, unpaired two-tailed Student's t-test, n = 5–6, *p < 0.05, vs. WT; #p < 0.05, vs. SREBP-1c S372A

metabolism through a post-translational modification. Metformin causes an AMPK-dependent activation of Insig through association and phosphorylation, which results in ablation of SREBP-1c proteolytic processing and activation, and attenuation of lipogenesis. Mechanistically, AMPK alpha subunit may directly phosphorylate Insig-1 at Thr222 site; and phosphorylation of Insig represses its ubiquitination and proteasomal degradation,

and leads to increased protein stability and activity of Insig. AMPK-dependent phosphorylation of Insig may represent a molecular mechanism by which AMPK agonists, such as metformin ameliorate hepatic steatosis.

One of the most important finding is that Insig-1 may be a novel substrate of AMPK. AMPK gain-of-function and loss-of-function approaches demonstrate that AMPK is sufficient and

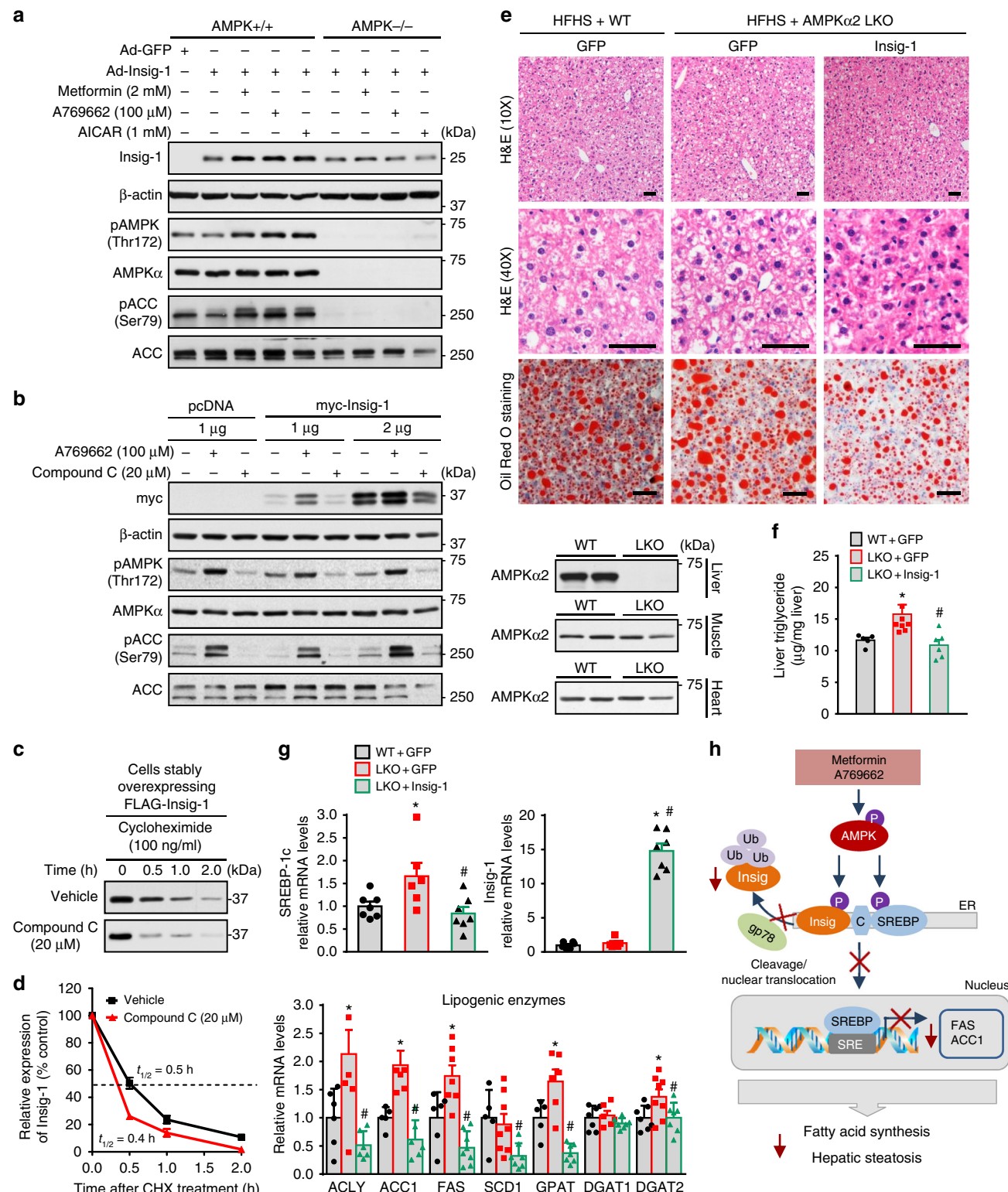

necessary for metformin, A769662 and AICAR on the stabilization and activation of Insig protein. Several evidences have demonstrated that Thr222 phosphorylation of Insig-1 is essential for AMPK to enhance Insig-1 activity and inhibit SREBP-1c proteolytic cleavage and target lipogenic gene expression. First, the phosphorylation of Insig-1 following AMPK activation was ablated by the nonphosphorylatable T222A mutant, indicating that Thr222 is specialized phosphorylated in an AMPK-

dependent manner. Second, the T222A mutant, not the others, eliminated metformin-caused induction of Insig-1, suggesting that Thr222 site is essential for AMPK's stimulating effects on Insig-1. Third, T222A mutant was resistant to A769662-caused protein stabilization, suggesting that AMPK-dependent phosphorylation at Thr222 site is essential for the stabilization of Insig-1. Fourth, T222A mutant diminished the inhibitory effects of A769662 on SREBP-1c cleavage. Moreover, the findings that

**Fig. 7** Hepatic overexpression of Insig-1 rescues hepatic steatosis in hepatocyte-specific AMPKα2 knockout mice fed with HFHS diet. **a–d** AMPK deficiency accelerates the degradation rates of Insig-1. **a** AMPK is required for Insig-1 induction in response to metformin, A769662 and AICAR. AMPK+/+ or AMPKα1/α2 double knockout (AMPK−/−) mouse primary hepatocytes were infected with adenoviruses encoding Insig-1 (Ad-Insig-1) or GFP for 24 h, followed by treatment with 2 mM metformin or AICAR for 24 h, or 100 μM A769662 for 8 h. **b** Compound C decreases Insig-1 protein levels in SRD-13A cells. Cells were transfected with pcDNA or plasmid encoding myc-tagged Insig-1 for twenty-four hours, followed by treatment with 100 μM A769662 or 20 μM compound C. **c, d** Compound C increases the degradation rate of Insig-1 in HEK293 cells stably expressing Insig-1. Cells were treated with 20 μM compound C for 24 h, followed by treatment with 100 ng/ml cycloheximide to inhibit protein synthesis as indicated. Data are presented as the mean ± SEM, $n = 4$. **e–g** Adenovirus-mediated overexpression Insig-1 attenuates hepatic steatosis in AMPKα2 LKO mice. Eight-week-old male WT and AMPKα2 LKO mice were fed with HFHS diet for 14 weeks, followed by treatment with adenoviruses encoding Insig-1 or control GFP by tail-vein injection for 2 weeks. **e** Representative H&E and oil red O staining of liver sections are shown. Scale bar, 50 μm. Expression levels of AMPKα2 in the liver, muscle or heart were measured by immunoblots. **f** Liver triglyceride levels were assessed. **g** Transcription levels of SREBP-1c and lipogenic enzymes are decreased in the liver of mice treated with Insig-1. mRNA levels of Insig-1, SREBP-1c and lipogenic enzymes are measured by real-time PCR. Data are presented as the mean ± SEM, unpaired two-tailed Student's $t$-test, $n = 6$–8, $*p < 0.05$, vs. WT and Ad-GFP; $\#p < 0.05$, vs. AMPKα2 LKO and Ad-Insig-1. **h** The proposed model for the post-translational regulation of lipogenesis via AMPK-mediated phosphorylation of Insig. AMPK phosphorylates Insig and represses its ubiquitination and degradation via inhibiting the interaction between Insig and the ubiquitin ligase gp78, which prevents the proteolytic processing and activation of SREBP-1c. The beneficial effects of metformin on hepatic steatosis are partially mediated by Insig

the physical association between AMPKα subunit and Insig-1 is enhanced by AMPK agonist A769662, suggest that activation of AMPK may enable its substrates to be more accessible to the kinase domain of AMPKα, which led us to propose that Insig-1 may be a direct substrate of AMPK. Together, these data suggest that AMPK, via Thr222 phosphorylation, inhibits proteasome-mediated degradation of Insig-1, which results in a reduction of SREBP-1c cleavage processing and lipogenic gene expression. Interestingly, the mutation of S46A blocks metformin-induced Insig-1 levels in Fig. 5e. It is possible that AMPK regulates Insig-1 at other phosphorylation sites. We proposed here that Insig-1 acts as a substrate of AMPK, however, it remains to be determined if this is a direct or indirect effect. Given that elevated protein levels and stability of Insig-2 are also increased by the treatment with AMPK agonists, it is conceivable that similar phosphorylation mechanism maybe involved. Further studies are needed to identify the AMPK phosphorylation site on Insig-2.

Here we found that nonphosphorylatable SREBP-1c S372A mutant remains responsive to metformin-inhibited proteolytic cleavage in HepG2 hepatocytes, although the magnitude is lowered than WT SREBP-1c protein. Taken together with AMPK-dependent phosphorylation and inhibition of SREBP[14], the current study provides a novel mechanism of AMPK in regulating fatty acid synthesis via Insig. The human HepG2 hepatocytes stably expressing SREBP-1c WT or S372A mutant protein are of importance for the study of SREBP proteolytic processing and activation.

The present study provides evidences that AMPK reduces proteasomal degradation of ubiquitinated Insig protein via gp78. First, AMPK activation by Metformin, A769662, or AICAR increases the protein levels of Insig and results in prolonged half-life. Second, AMPK represses ubiquitination and degradation of both Insig isoforms. Compared with Insig-2, Insig-1 is highly ubiquitinated, which is consistent with previous observation showing a much more rapid degradation of Insig-1[10]. Moreover, activation of AMPK reverses the gp78-mediated degradation of Insig-1 likely via disrupting association between Insig and gp78. These data suggest that AMPK increases Insig stability by inhibiting gp78-mediated proteasome pathway. Proteasomal degradation of ubiquitinated protein is widely regulated by a phosphorylation-dependent mechanism. Phosphorylation of LIN28, a highly conserved RNA-binding protein, by MAPK/ERK increases its protein stability[27]. The proinflammatory kinase Inhibitor of nuclear factor κB kinase β (IKKβ) phosphorylates X-box-binding protein 1 (XBP1) inhibits its protein degradation mediated by ubiquitin proteasome pathway[28]. Interestingly,

AMPK-dependent phosphorylation of thioredoxin-interacting protein (TXNIP) leads to its accelerated degradation and improved glucose uptake[29]. We report here that AMPK physically associates with Insig-1 to enhance Insig-1 protein stability via phosphorylation.

Recent studies demonstrated that the transcription of Insig-2a is regulated during the adaptive metabolic response to fasting or refeeding. During fasting, Insig-2a is induced by glucocorticoid-SETDB2 signaling[30], whereas during refeeding, Insig-2a is potently inhibited by a novel metabolic regulator CREBZF[15]. Together with the current findings showing post-translational regulation of Insig via AMPK, the dynamic regulation of Insig provides a finely tuned mechanism for the cellular regulation of hepatic lipid homeostasis.

Metformin has long been recommended as first-line treatment in various guidelines for treating type 2 diabetes. Treatment with metformin has been shown to improve NAFLD phenotypes in genetically obese ob/ob mice[31], and in humans[16–19]. We report here that metformin protects against hepatic steatosis in DIO mice. Beneficial effects of metformin on hepatic steatosis are at least partially mediated through AMPK-Insig pathway. First, metformin stimulates AMPK activity in the liver of HFHS diet-fed mice, which is positively correlated with increased expression levels of Insig-1 and Insig-2. Second, metformin increases the half-life of Insig-1 protein. Third, treatment with metformin reverses the gp78-mediated degradation of both Insig isoforms. Moreover, SREBP-1c S372A mutant remains responsive to metformin-caused reduction of SREBP-1c proteolytic activation and lipid accumulation in HepG2 cells, suggesting a potential role of the novel AMPK substrate Insig in mediating metformin's effects in lipogenesis. These results identify Insig as a novel mediator in regulating metformin's beneficial effects on hepatic lipogenesis and lipid metabolism. Further studies are needed to validate these mechanisms in humans.

The in vivo and in vitro studies demonstrate that augmentation of Insig activity is efficient to inhibit hepatic lipogenesis and attenuate excessive lipid accumulation in hepatocytes and in livers. Insig activation by AMPK inhibits the cleavage of SREBP-1c and reduces lipogenic gene expression and lipid accumulation. Moreover, adenovirus-mediated overexpression of Insig-1 rescues hepatic steatosis in AMPKα2 LKO mice fed with HFHS diet, which is consistent with the reduced hepatic lipogenesis and triglyceride levels in the liver-specific Insig-1 transgenic mice[8]. In this study, moderate hepatic steatosis was observed in AMPKα2-deficient mice fed with HFHS diet, whereas previous reports showed that liver-specific AMPKα2 knockout mice fed with chow

diet develop glucose intolerance in response to fasting without obvious hepatic steatosis[32]. The difference may be attributed to the different components of the diets, different pathophysiological conditions or different genetic backgrounds of the mice. Nevertheless, it would be interesting in future studies to address these questions. Together, these results support the notion that Insig plays a central role in regulating lipogenic gene expression and lipogenesis[9,15,33,34], and that Insig may serve as a therapeutic target to ameliorate hepatic steatosis and related metabolic disorders.

In summary, the current study identifies a biochemical mechanism of Insig regulation by AMPK (Fig. 7h). AMPK activation stabilizes Insig protein via phosphorylation. Pharmacological and genetic approaches for the modulation of Insig activity may provide novel therapeutic avenues for the treatment of metabolic diseases, such as fatty liver disease, insulin resistance, and type 2 diabetes.

## Methods

**Animal model and diets.** Liver-specific AMPKα1/α2 double knockout (AMPK −/−) mice were generated by crossing floxed AMPKα1/α2 mice, which was generated by crossing floxed AMPKα1 mice with floxed AMPKα2 mice (stock number: 014141 and 014142, respectively, Jackson Laboratory, ME), with albumin-cyclization recombination (Alb-Cre) recombinase transgenic mice (Model Animal Research Center of Nanjing University, China); WT littermates (Alb-Cre-negative, AMPKα1/α2$^{flox/flox}$) were used as the control. Liver-specific AMPKα2 knockout (AMPKα2 LKO) mice were generated by crossing the floxed AMPKα2 mice with Alb-Cre recombinase transgenic mice, the WT littermates (albumin-Cre negative, AMPKα2$^{flox/flox}$) were used as the control. Male C57BL/6 mice at 8 weeks of age were purchased from Shanghai Laboratory Animal Co. Ltd, China. Mice were placed on high-fat and high-sucrose (HFHS) diet (D12327, Research Diets)[35,36] for 8 weeks, and then fed with HFHS diet supplemented with daily metformin (50 mg/kg/day) or vehicle (PBS) treatment via intraperitoneal injection for 8 weeks. All mice were housed under a 12:12-h light/dark cycle at controlled temperature. All animal experimental protocols were approved by Institutional Animal Care and Use Committee at Shanghai Institute of Nutrition and Health, Shanghai Institutes for Biological Sciences, Chinese Academy of Sciences.

**Liver histological analysis.** Livers were fixed in 10% phosphate-buffered formalin acetate at 4 °C overnight and embedded in paraffin wax. Paraffin sections (5 μm) were cut and mounted on glass slides for H&E staining[35,36]. Immunohistochemistry of liver sections was performed[37]. Livers embedded in optimum cutting temperature compound (Tissue-Tek, Laborimpex) were used for oil red O staining for the assessment of hepatic steatosis according to the manufacturer's instructions (American MasterTech, Lodi, CA).

**In vivo adenoviral gene transfer.** Adenovirus-mediated gene transfer of Ad-Insig-1 or Ad-GFP in the liver of AMPKα2 LKO mice or WT littermates was performed[35]. Briefly, adenoviruses ($5 \times 10^9$–$1 \times 10^{10}$ pfu per mouse) were delivered into mice by tail-vein injection. Two weeks post-injection, the mice were sacrificed in a postprandial state under isoflurane anesthesia, and tissues were rapidly taken and freshly frozen in liquid nitrogen and then stored at −80 °C or fixed for histologic analysis.

**Lentivirus production, infection, and selection.** Lentiviral particles were generated[35,38]. Briefly, HEK293T cells were transfected with lentiviral transfer plasmid (pCDH-CMV-3xFLAG-Insig-1, pCDH-CMV-FLAG-SREBP-1c, or pCDH-CMV-FLAG-SREBP-1c S372A), along with packaging plasmids pMDLg/pRRE and pRSV-Rev (1:1 ratio), and envelope plasmid pMD2.G using polyethylenimine (PEI). The media containing lentiviral particles was stored at 4 °C. For lentivirus infection, HEK293 or HepG2 cells were cultured, treated with lentiviral particle solution, and then selected and passaged under puromycin-containing media for 4–7 days. The resulting cells stably expressing Insig or SREBP were ready for further assays.

**Reagents and antibodies.** AMPK agonist metformin (cat. D150959), cyclohex-imide (cat. C1988), Oil Red O (cat. O0625), and mouse monoclonal FLAG M2 antibody (cat. F3165) were purchased from Sigma-Aldrich (St. Louis, MO). A769662 (cat. sc-203790) was from Santa Cruz Biotechnology (Santa Cruz, CA). AICAR (cat. A611700) was from Toronto Research Chemicals. Compound C (cat. 11967) was from Cayman Chemical (Ann Arbor, MI). MG132 (cat. 1748) was from TOCRIS Bioscience (Bristol, UK). Glutathione-Sepharose beads (cat. 17-0756-01) were from GE Healthcare (Piscataway, NJ). Rabbit monoclonal phosphor-AMPKα (Thr172) antibody (1:1000, cat. 2535), rabbit monoclonal total AMPKα pan antibody (1:1000, cat. 2532), rabbit polyclonal Histone H3 antibody (1:1000, cat. 9715),

rabbit monoclonal GAPDH antibody (1:2000, cat. 5174), mouse monoclonal myc-tag antibody (1:1000, cat. 2276) and rabbit monoclonal HA-Tag antibody (1:1000, cat. 3724) were obtained from Cell Signaling Technology (Beverly, MA). Rabbit polyclonal antibody against phospho-acetyl CoA carboxylase (Ser79) (1:1000, Cat. 07-303) was obtained from Merck Millipore. Total acetyl CoA carboxylase (1:1000, Cat. 3662) was obtained from Upstate Biotechnology (Lake Placid, NY). Mouse monoclonal FAS antibody (1:2000, cat. 610963), mouse monoclonal phosphoser-ine/threonine (pSer/Thr) antibody (1:1000, cat. 612549) were from BD Biosciences (San Jose, CA). Antibodies against Insig-1 (1:500, cat. sc-390504), Insig-2 (1:500, cat. sc-66936), GST (1:1000, cat. sc-138), SREBP-1 (1:1000, K10, sc-367), β-actin (1:2000, cat. sc-69879); horseradish peroxidase-conjugated anti-mouse (1:10,000) and anti-rabbit (1:10,000) secondary antibodies; and protein A/G PLUS-Agarose beads were obtained from Santa Cruz Biotechnology (Santa Cruz, CA). The antibody against SREBP-2 (1:1000, cat. ab30682) was purchased from Abcam (Cambridge, MA).

**Plasmid construction and transfection.** The plasmid encoding FLAG-tagged human Insig-2 was constructed by amplifying PCR products from pCMV-Insig-2-myc (ATCC, VA, USA) and inserting it into the EcoRI/XhoI site of FLAG-tagged pcDNA vector. The following primers were used: AATGCGAATTCATGGCAG AAGGAGAGACAGA (F) and AATGCCTCGAGTCATTCCTGATGAGATTTTT (R). The plasmid encoding HA-Ub has been described previously[39]. To generate the plasmids encoding nonphosphorylatable S43A, S46A, S74A, S125A, S178A, S190A, S209A, T216A, T222A, T238A, S276A mutants of Insig-1, site-directed mutagenesis was carried out on myc-Insig-1. The following primers were used: for S43A mutant, AACGTTTCCGTGGCCGGGCCCTCCC (F) and GGGGAGGGGCC CGGCCACGGAAACGTT (R); for S46A mutant, TCCGGGCCCGCCCTGCTG GCG (F) and CGCCAGCAGGGCGGGCCCGGACA (R); for S74A mutant, CC CGAGCCCGGCCGCCCCTACCCCAA (F) and TTGGGGTAGGGGGCGCCGGG GCTCGGG (R); for S125A mutant, CCATCTTTTCGCCGCCTGGTGGGTCC (F) and GGACCCACCAGGCGGCGGAAAAGATGG (R); for S178A mutant, TA ACCACGCCGCTGCTAAATTGG (F) and CAATTTAGCAGCGGCGTGGTTAA TGC (R); for the S190A mutant, TGTCCAGCTGGCCTTGACTTTAGC (F) and TGCTAAAGTCAAGGCCAGCTGGAC (R); for S209A mutant, GATCGTTCC AGAGCTGGCCTTGG (F) and CCAAGGCCAGCTCTGGAACGATC (R); for T216A mutant, GCTAGAAAAGCTATGGCGATCCCCAGCCCAAGG (F) and CCTTGGGCTGGGGATCGCCATAGCTTTTCTAGC (R); for T222A mutant, GCGTGATCAGCGCAGCTAGAAAAGCTATGGTGATCC (F) and GGATCAC-CATAGCTTTTCTAGCTGCGCTGATCACGC (R); for T238A mutant, CTATC AGTATACAGCCACAGATTTC (F) and AGAGGAAATCTGGGGCTGTATAC TG (R); for T276A mutant, GTTCCTGAAAAGCCCCATGCTGATGCGGCCG (F) and CGGCCGCCATCAGCATGGGGCTTTTCAGGAAC (R). Mutated bases were indicated by underlining. All constructs and mutations were verified by DNA sequencing. Transfection assays for plasmids were performed using Lipofectamine 2000 (Life Technologies) according to the manufacturer's protocol.

**Nuclear and cytoplasmic extraction.** Nuclear and cytosolic extraction was performed using the NE-PER nuclear and cytoplasmic extraction kit (cat. 78833) obtained from Thermo Fisher Scientific. Briefly, HepG2 cells were treated by metformin for 24 h, harvested with trypsin–EDTA and then centrifuged at 500×g for 5 min. Next, the cell pellets were washed with PBS, and centrifuged at 500×g for 5 min. To get the cytoplasmic extracts, the supernatant was removed and discarded, 200 μl ice-cold CER I was added to the cell pellet. The tube was vortexed vigorously on the highest setting for 15 s, then the tube was incubated on ice for 10 min. Eleven microliters of ice-cold CER II was added to the tube, followed by vortex and centrifuged at 16,000×g for 5 min. The supernatant containing the cytoplasmic fraction was transferred to a new tube. To get the nuclear extract, the pellet fraction was suspended in 100 μl ice-cold NER, the tube was vortexed vigorously on the highest setting for 15 s, and the sample was placed on ice and vortexing was continued for 15 s every 10 min for a total of 40 min. After last centrifuge at 16,000×g for 5 min, the supernatant containing the nuclear fraction was transfered to a new tube. The cytoplasmic and nuclear extracts were collected and used for Immunoblots.

**LC–MS/MS analysis.** Cell lysates prepared from myc-tagged Insig-1-transfected HEK293 cells were incubated with anti-myc-conjugated beads. The immunopre-cipitated proteins were resolved by SDS–PAGE and stained with Coomassie Brilliant Blue dyes. Protein bands that corresponds to myc-Insig-1 from A769662-treated or untreated cells were excised and digested with trypsin buffer for 16 h at 37 °C. The resultant peptides were analyzed on an ultra-high performance liquid chromatography system (AMR) coupled to an Orbitrap Q Exactive mass spectrometer (Thermo Fisher Scientific). To generate an extracted ion chromatogram, the raw data were processed using Sequest and Proteome Discoverer (Thermo Scientific) software and directly analyzed using Proteome Discoverer (Beijing Bangfei Bioscience, China).

**Immunoblots, Co-IP, and GST pull-down.** Immunoblotting analysis was carried out[14,36]. In brief, cells were homogenized and lysed at 4 °C in lysis buffer (50 mM Tris–HCl, pH 8.0, 1% (v/v) Nonidet P-40, 150 mM NaCl, 5 mM EDTA, 1 mM

EGTA, 1 mM sodium orthovanadate, 10 mM sodium fluoride, 1 mM phe-nylmethylsulfonyl fluoride, 2 µg/ml aprotinin, 5 µg/ml leupeptin, and 1 µg/ml pepstatin). Cell lysates were centrifuged at 16,000 ×g for 10 min at 4 °C, and the resulting supernatant was used for immunoblotting analysis. Protein concentrations in cell lysates were measured using Bio-Rad Protein Assay Dye Reagent. For immunoblotting, 20–50 µg of protein was separated by 8–10% sodium dodecyl sulfate–polyacrylamide gel electrophoresis (SDS–PAGE), and then electrophoretically transferred to polyvinylidene difluoride (PVDF) membranes in a transfer buffer consisting of 25 mM Tris base, 190 mM glycine, and 20% methanol. The membranes were blocked with 5% non-fat milk in Tris-buffered saline with 0.1% Tween 20 (TBST) and incubated with specific antibodies, followed by incubation with horseradish peroxidase-conjugated secondary antibodies. Immunoblots were visualized by LumiGLO chemiluminescence detection kit (Cell Signaling Technology). The intensity of bands was quantified using ImageJ (National Institutes of Health, Bethesda, MD). Relative phosphorylation levels were normalized to those of endogenous proteins and presented as the means ± SEM. For Co-IP, cell lysates containing overexpressed recombinant or endogenous proteins were incubated with specific antibodies and protein A/G Sepharose beads at 4 °C for overnight. The precipitates were washed three times with ice-cold lysis buffer. For GST pull-down analysis, cell lysates containing overexpressed glutathione S-transferase (GST) tagged fusion proteins or endogenous proteins were incubated with Glutathione Sepharose 4B beads (GE Healthcare Bio-Sciences, PA) at 4 °C overnight and washed as Co-IP. The precipitates were then analyzed by immunoblots.

**Primary mouse hepatocyte isolation and culture.** Primary mouse hepatocytes were isolated[35]. Briefly, six-well plates were coated by rat-tail collagen type I (cat. 08-115, Millipore) for overnight, and then washed by PBS for twice. Briefly, mice were anesthetized with sodium pentobarbital (30 mg/kg intraperitoneally), and the portal vein was cannulated under aseptic conditions. The liver was perfused with ethylene glycol-bis (2-aminoethylether)-N,N,N′,N′-tetraacetic acid (EGTA) solution (5.4 mmol/l KCl, 0.44 mmol/l KH$_2$PO$_4$, 140 mmol/l NaCl, 0.34 mmol/l Na$_2$HPO$_4$, 0.5 mmol/l EGTA, 25 mmol/l Tricine, pH 7.2) and Hank's balanced salt solution (HBSS) containing 0.075% collagenase type I (Sigma-Aldrich), 10 mg/ml DNase I (Sigma-Aldrich), 200 units/ml penicillin, and 200 µg/ml streptomycin, and then digested with 0.025% collagenase solution for the mouse liver. The isolated mouse hepatocytes were then cultured at 80–90% confluence in DMEM medium containing 10% FBS in rat-tail collagen type I-coated six-well plates (BD Biosciences) for overnight. Cells were infected with adenoviruses expressing Insig-1 or GFP, treated without or with metformin, A769662 and AICAR in a serum-free medium.

**Cell lines and cell treatment.** Human HepG2 hepatocyte and HEK293 cells were purchased from Cell Bank, Type Culture Collection Committee, Chinese Academy of Sciences (Shanghai, China). The cells were cultured, and treated as indicated[14,35,40]. Cells were cultured in DMEM containing 5.5 mM D-glucose,10% fetal bovine serum, 100 units/ml penicillin, and 100 µg/ml streptomycin, and incubated in a humidified atmosphere of 5% CO$_2$ at 37 °C and passaged every 2 days by trypsinization. Cells were transfected with plasmids for 24 h, followed by treatment with 2 mM metformin for 24 h, 5 µM MG132 for 8 h, or 1 mM AICAR 24 h. For adenovirus infection, cells were treated with adenoviruses encoding constitutive active AMPK (CA-AMPK) or GFP for 48 h. SRD-13A cells (deficient in SCAP derived from CHO-7 cells)[7] were generous gifts from Drs. Michael S. Brown and Joseph L. Goldstein at University of Texas Southwestern Medical Center, USA. The cells were grown in monolayer at 37 °C in 5% CO$_2$, and were maintained in medium D. Medium D including 1:1 mixture of Ham's F12 medium and Dulbecco's modified Eagle's medium containing 100 units/ml penicillin, and 100 µg/ml streptomycin, supplied 5% fetal bovine serum, 5 µg/ml cholesterol, 1 mM sodium mevalonate, and 20 µM sodium oleate[41].

**RNA isolation and real-time PCR analysis.** Liver tissues or cells were lysed in TRIzol Reagent (Life Technologies) to extract total RNAs according to the manufacturer's protocol. Total RNAs were then reversely transcribed to cDNA using SuperScript II reverse transcriptase (Life Technologies) and Oligo d (T). The resulting cDNA was subjected to real-time PCR with gene-specific primers in the presence of SYBR Green PCR master mix (Applied Biosystems) using StepOnePlus Real-Time PCR System (Applied Biosystems)[15].

**Statistical analysis.** Data are expressed as mean ± SEM. Statistical significance was evaluated using the unpaired two-tailed Student's t-test and among more than two groups by analysis of one-way ANOVA. Differences were considered significant at a P-value < 0.05.

## Data availability
The authors declare that all data supporting the findings of this investigation are available within the article, its Supplementary Information, and from the corresponding authors upon reasonable request.

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

## Acknowledgements

This work was supported by grants from National Key R&D Program of China (2017YFC0909601), National Natural Science Foundation of China (31471129 and 31671224), and Chinese Academy of Sciences (ZDBS-SSW-DQC-02) to Y.L. This work was also supported by grants from National Natural Science Foundation of China (81125023) to Jia.L. and K.C. Wong Education Foundation to Jia.L. and Y.L. We are grateful to Drs. Michael S. Brown and Joseph L. Goldstein (University of Texas Southwestern Medical Center) for providing myc-tagged Insig-1 plasmid and SRD-13A cells, Dr. Chen Wang (SIBS, China) for providing myc-tagged gp78 plasmid and Dr. Zhonghui Weng for technical assistance.

## Author contributions

Y.H., Z.H. and Y.L. contributed to experiment design; Y.H., Z.H., A.C., Z.L., Y.X., Y.L., F.M., F.Z., Z.Z., J.G. contributed to the acquisition and analysis of data; Y.Y., Jin.L., Jia.L., J.F., B.S. provided reagents and material support; C.W., J.F. and J.-G.F. reviewed the manuscript; Y.L. and Jia.L. obtained the funding; Y.H., Z.H. and Y.L. wrote the manuscript.

## Additional information

**Competing interests:** The authors declare no competing interests.

