## [Peer Review File · Nature Communications]

Reviewers' comments:

Reviewer #1 (Remarks to the Author):

In this manuscript, the authors report that activation of AMPK leads to phosphorylation of Insig-1 and Insig-2. This phosphorylation of Insig results in reduced affinity of the proteins for the E3 ubiquitin ligase gp78. In the absence of gp78 association, Insig proteins accumulate and inhibit proteolytic processing of SREBPs, thereby explaining reduced lipid accumulation in animal models treated with AMPK activators. Overall, the conclusion of the manuscript are somewhat supported by immunoblot assays, Insig ubiquitination assays, and co-immunoprecipitations. However, the results are correlative and experiments demonstrating AMPK directly phosphorylates Insigs are not provided. The authors need to identify the site of AMPK-dependent phosphorylation in Insig-1 and Insig-2. This site should be mutated and experiments provided showing their mutations prevent AMPK-mediated phosphorylation and down-regulation of SREBP processing. Conversely, authors should generate a phosphomimetic mutation as well.

Reviewer #2 (Remarks to the Author):

In Han and Hu et al., the authors set out to investigate the regulation of SREBP processing by AMPK. They had previously identified a role for direct AMPK phosphorylation of SREBP as a mechanism by which AMPK inhibits lipogenesis. In this study, they extend on that work and claim that direct phosphorylation of Insig-1 and Insig-2, which are critical regulators of SREBP processing and activation, may be another mechanism by which AMPK can regulate lipid metabolism. Although regulation of Insig by AMPK agonists is evaluated to a large extent (except for a few key points – see below), the work presented here falls short from rigorously demonstrating that Insigs are directly phosphorylated by AMPK, especially in the absence of an identified site. Nevertheless, the data presented on the regulation of Insig stability by AMPK is interesting and, provided the specific points below are addressed, could be considered for publication at Nature Communications without claiming direct phosphorylation of Insig.

Major points:

1. Figure 5 is supposed to establish Insig-1 as a direct AMPK substrate. The evidence for this is weak and does not meet the standard required to claim that AMPK directly phosphorylates Insig. First, the previous 3 figures have clearly and unambiguously demonstrated that AMPK activation leads to Insig stabilization. Therefore, the observation that AMPK-Insig binding increases after AMPK activation, as well as the increased reactivity of the pSer/Thr antibody merely reflects the higher level of Insig protein resulting from AMPK activation. Similarly, Fig5D shows that compound C (a very non-specific inhibitor) decreases both the flag signal as well as the pSer/Thr signal on the IP blot. Unless the authors are able to identify a specific site that is phosphorylated by AMPK, the data presented here do not support the claim that AMPK directly phosphorylates Insig and the text should be edited to reflect the fact that AMPK regulates Insig stability and interaction with gp78 by an unidentified mechanism. DUBs, E3 ligases, all kinds of potential direct substrates of AMPK could be controlling Insig turnover without invoking direct phosphorylation of Insig without compelling data.

2. Figure 1 clearly demonstrates that the levels of Insig1 and 2 are increased in response to metformin, although not necessarily through AMPK. Figure 2 attempts to demonstrate that this happens via AMPK by using CA-AMPK (and A769662 in other figures). However, this is done only in HEK293 cells that overexpress Insig-1 and 2. The authors should demonstrate that specific AMPK activators lead to increases in Insigs in either the liver (A769662 should activate AMPK in vivo) or at least in more physiologically relevant cells (hepatocytes, preferably, or HepG2) and include those data on Figure 2. Appropriate controls, such as AMPK KO cells or siRNA against AMPK should be used.

Related to this, on page 6, first paragraph, the authors claim that “these data indicate that AMPK is sufficient to stimulate both Insig-1 and Insig-2 activity in vivo...” Again, without a demonstration that specific AMPK activation in the liver (for example A769662 treatment in mice) increases Insig, the authors should not make this “in vivo” claim.

3. In Figure 4, Metformin is shown to decrease Insig-gp78 interaction. However, the treatment time is rather long (8h) for a direct effect of AMPK on this interaction. Is this treatment time required for the effect? Metformin is likely to activate AMPK within 1-2h of treatment.

4. Figure 6 evaluates how SREBP processing is affected by AMPK activity. The authors do this by looking at precursor and nuclear SREBP1 that is overexpressed in HepG2 cells. This figure would benefit greatly if the authors instead performed cellular fractionation to look at this. This is especially true in blots where both the precursor and the nuclear SREBP1 go down (in figure 6D, for example), where it is harder to be confident that processing was inhibited (rather than simply the total levels going down). In addition, nuclear fractionation would allow detection of endogenous nuclear SREBP. The authors should do an experiment, such as the one on figure 6A, with cellular fractionation and preferably looking at endogenous SREBP1 and Insig. Additionally, the quantification shown in Fig 6C should analyze the ratio of nSREBP/pSREBP to normalize for variation in expression level.

5. In figure 6F, the authors claim that the very last picture has less Oil Red O than the previous picture (metformin-treated SREBP-1c-S372A cells have less lipids than vehicle control). This is not obvious at all from the picture. Please quantify the Oil Red O staining, so the result can be visualized better.

6. In Figure 7, the authors show that Insig-1 overexpression in AMPKa2 LKO rescues some features of steatosis. However, this experiment is somewhat misleading, as increasing Insig expression is likely to correct liver steatosis in WT as well (which is not shown here). Therefore, the connection to the specific phenotype of AMPKa2 LKO is hard to understand. More should be done to show that Insig1 is necessary for the effect of Metformin and direct AMPK activators on SREBP processing. For instance, CRISPR KO cell lines for AMPK and Insig could be generated in HepG2 and their contribution to the effect of Metformin on SREBP processing as well as on lipogenic gene expression should be evaluated (similarly to Figure 7G). Alternatively, if the authors have access to AMPK-null liver (AMPKa2 LKO likely still have AMPKa1 and residual AMPK activity) they could evaluate the effect of Metformin on Insig stabilization in the absence of AMPK, which would really establish AMPK as required for this phenomenon in vivo.

Minor points:

1. On page 4, line 1: the authors say “We have previously identified that AMPK is a direct upstream kinase of SREBP. AMPK phosphorylation of SREBP-1c at ser372 site is sufficient and required for the inhibition of proteolytic cleavage and nuclear translocation of SREBP-1c15. However, SREBP-1c S372A mutation remains responsive to AMPK-mediated proteolytic cleavage and maturation of SREBP-1c, albeit the extent is less than wild-type (WT) SREBP-1c.”

This sounds contradictory. First sentence says that AMPK inhibits cleavage, but the second sentence says “AMPK-mediated proteolytic cleavage”. Is it not the model that AMPK inhibits cleavage? What do the authors mean by “AMPK-mediated cleavage”? Similar comment for page 4, second paragraph: “AMPK-induced proteolytic cleavage in hepatocytes”. Please clarify.

2. Page 4, last paragraph. The authors should not refer to their metformin treatment as “acute”. 8 weeks of metformin on diet is best described as “sustained”.

3. Page 7, last paragraph: “kinase assay”. The authors did not perform a kinase assay, but simply a western blot. Please correct.

4. page 8, line 1. The authors say "... leads to an augmentation of Insig-1 activity and reduction of fatty acid synthesis". The authors have not evaluated Insig-1 activity or fatty acid synthesis at this point in the paper. They do later, so a claim like that should appear in the text only after the data on Figures 6 and 7 are discussed.

5. Typo: page 4, second paragraph: change "has" to "have".

6. Typo: page 5, line 1: change "straining" to "staining".

7. Typo: page 5, paragraph 3: change "metformin were" to "metformin was".

8. Typo: page 7, end of page: "were evidence". Change. Do they mean "evident"? Evidence of what?

9. Typo: page 9: change "attenuate" to "attenuates"

10. Typo: page 11, first paragraph: The sentence "To test the hypothesis that activation of AMPK..." is either incomplete (it doesn't say what will be done to test the hypothesis) or just wrong. Please change.

Reviewer #3 (Remarks to the Author):

Han and colleagues have investigated a possible new mode of SREBP cleavage regulation via AMPK-mediated phosphorylation of Insig. They first show in a DIO model that chronic administration of metformin results in increased Insig-1 and -2 protein levels. This was associated with reduced liver TG, cholesterol and FAS expression. They then turn to in vitro studies using transfection to show AMPK activation leads to increased Insig-1 and -2 protein levels. Further studies show that ubiquitination of Insig is reduced by AMPK activation as is protein degradation. Overexpression studies also suggested that AMPK activation partially attenuates the known role of gp78 in Insig degradation. Co-IP studies again with overexpressed protein suggest that AMPK and Insig directly interact and an activator of AMPK increased Insig phosphorylation. TO assess functional Insig, cells stably transfected with full length SREBP-1c were treated with metformin, which reduced nuclear SREBP-1 protein levels. Finally, Insig overexpression in AMPKa2 KO mice reduced liver TGs.

Major Points:

1. Fig. 1 needs nuclear SREBP-1 and SREBP-2 measurements.
2. Fig. S1- Although not entirely clear, it appears the 293 cells were transfected with a plasmid encoding Insig and then treated with metformin. The mRNA for Insig did not change with treatment so the conclusion is that metformin does not change the mRNA levels despite the increase in protein levels. This is true in the transfected state but one would predict that endogenous mRNA levels of Insig1 would decrease since nuclear SREBP-1 should be reduced.
3. The final in vivo study is not proof that lack of Insig is responsible for the phenotype. It is likely that Insig overexpression in WT mice fed the HFHS diet would also reduce liver TGs.
4. The studies would be more convincing if the phosphorylation site(s) were identified and mutants studied.

Minor Points:

1. In general the abstract is not very clear for the average reader and the first and second sentences in particular are too confusing.
2. It is repeatedly stated there are clinical studies that indicate metformin may be beneficial for humans with NASH- actually all of the studies suggest there is no significant benefit for NASH and

no further trials are ongoing.

Reviewer #1: *In this manuscript, the authors report that activation of AMPK leads to phosphorylation of Insig-1 and Insig-2. This phosphorylation of Insig results in reduced affinity of the proteins for the E3 ubiquitin ligase gp78. In the absence of gp78 association, Insig proteins accumulate and inhibit proteolytic processing of SREBPs, thereby explaining reduced lipid accumulation in animal models treated with AMPK activators. Overall, the conclusion of the manuscript are somewhat supported by immunoblot assays, Insig ubiquitination assays, and co-immunoprecipitations. However, the results are correlative and experiments demonstrating AMPK directly phosphorylates Insigs are not provided. The authors need to identify the site of AMPK-dependent phosphorylation in Insig-1 and Insig-2. This site should be mutated and experiments provided showing their mutations prevent AMPK-mediated phosphorylation and down-regulation of SREBP processing. Conversely, authors should generate a phosphomimetic mutation as well.*

Answer: We would like to thank Reviewer#1 for thoughtful and constructive comments. First, to identify AMPK-dependent phosphorylation sites in Insig-1, a mass spectrometry proteomic analysis was performed. Six putative AMPK sites, including Ser43, Ser46, Ser74, Thr216, Thr222 and Ser276, were identified within the protein sequence of Insig-1 (Fig. S3A). The mutation of T222A, not the others, eliminated metformin-caused induction of Insig-1 (Fig. 5E and S3B), suggesting that T222 site is essential for AMPK's stimulating effects on Insig-1. The phosphorylation of Insig-1 by AMPK was ablated by the nonphosphorylatable T222A mutant (Fig. 5F), indicating that T222 is specialized phosphorylated by AMPK. Second, as shown in Fig. S3C and Fig. 5G, compared with cells treated with vehicle, the proteasome-mediated degradation of WT Insig-1 and S74A mutant in the presence of A769662 were significantly reduced, whereas T222A mutant was resistant to A769662-caused protein stabilization. These data suggest that AMPK phosphorylation at T222 site is essential for the stabilization of Insig-1. Third, as shown in Fig. 6E, the amount of nuclear form of SREBP-1c derived from HepG2 cells stably expressing FLAG-tagged SREBP-1c S372A was largely reduced by WT Insig-1 when treated with A769662, accompanied by an reduction of the lipogenic enzyme FAS. Conversely, the T222A mutant diminished the inhibitory effects of A769662 on SREBP-1c cleavage and expression of FAS. Together, these data suggest that AMPK, via T222 phosphorylation, inhibits proteasome-mediated degradation of Insig-1, which results in a reduction of SREBP-1c cleavage processing and lipogenic gene expression.

Reviewer #2:

In Han and Hu et al., the authors set out to investigate the regulation of SREBP processing by AMPK. They had previously identified a role for direct AMPK phosphorylation of SREBP as a mechanism by which AMPK inhibits lipogenesis. In this study, they extend on that work and claim that direct phosphorylation of Insig-1 and Insig-2, which are critical regulators of SREBP processing and activation, may be another mechanism by which AMPK can regulate

lipid metabolism. Although regulation of Insig by AMPK agonists is evaluated to a large extent (except for a few key points – see below), the work presented here falls short from rigorously demonstrating that Insig is directly phosphorylated by AMPK, especially in the absence of an identified site. Nevertheless, the data presented on the regulation of Insig stability by AMPK is interesting and, provided the specific point below are addressed, could be considered for publication at Nature Communications without claiming direct phosphorylation of Insig.

Answer: We would like to thank Reviewer#2 for thoughtful and positive comments. We have now revised our manuscript according to the comments as detailed below.

Major points:

1. Figure 5 is supposed to establish Insig-1 as a direct AMPK substrate. The evidence for this is weak and do not meet the standard required to claim that AMPK directly phosphorylates Insig. First, the previous 3 figures have clearly and unambiguously demonstrated that AMPK activation leads to Insig stabilization. Therefore, the observation that AMPK-Insig binding increases after AMPK activation, as well as the increased reactivity of the pSer/Thr antibody merely reflects the higher level of Insig protein resulting from AMPK activation. Similarly, Fig5D shows that compound C (a very non-specific inhibitor) decreases both the flag signal as well as the pSer/Thr signal on the IP blot. Unless the authors are able to identify a specific site that is phosphorylated by AMPK, the data presented here do not support the claim that AMPK directly phosphorylate Insig and the text should be edited to reflect the fact that AMPK regulates Insig stability and interaction with gp78 by an

unidentified mechanism. DUBs, E3 ligases, all kinds of potential direct substrates of AMPK could be controlling Insig turnover without invoking direct phosphorylation of Insig without compelling data.

Answer: We would like thank Reviewer#2 for thoughtful and constructive comments. Through integrative mass spectrometry proteomic and genetic analysis, we identified that Thr222 phosphorylation of Insig-1 by AMPK is necessary for augmentation of Insig-1 activity and its inhibitory effects on SREBP-1c cleavage and lipogenic gene expression. Please see the same answers to the question also raised by Reviewer#1.

2. Figure 1 clearly demonstrates that the levels of Insig1 and 2 are increased in response to metformin, although not necessarily through AMPK. Figure 2 attempts to demonstrate that this happens via AMPK by using CA-AMPK (and A769662 in other figures). However, this is done only in HEK293 cells that overexpress Insig-1 and 2. The authors should demonstrate that specific AMPK activators lead to increases in Insig in either the liver (A769662 should activate AMPK in vivo) or at least in more physiologically relevant cells (hepatocytes, preferably, or HepG2) and include those data on Figure 2. Appropriate controls, such as AMPK KO cells or siRNA against AMPK should be used.

Related to this, on page 6, first paragraph, the authors claim that “these data indicate that AMPK is sufficient to stimulate both Insig-1 and Insig-2 activity in vivo...” Again, without a demonstration that specific AMPK activation in the liver (for example A769662 treatment in mice) increases Insig, the authors should not make this “in vivo” claim.

Answer: Thanks for the insightful comments. As shown in revised Fig. 2C and 2F, HepG2 cells, instead of HEK293 cells, were infected with adenoviruses encoding constitutive active AMPK (CA-AMPK) or GFP, and then followed by immunoblotting analysis. These data suggest that constitutive activation of AMPK is sufficient to enhance the protein levels of Insig-1 and -2 in hepatocytes. We have revised the manuscript accordingly. To further characterize that AMPK is required for the stabilization and activation of Insig-1 protein, we found that protein levels of Insig-1 was increased by treatment with AMPK activators including metformin, A769662 and AICAR in AMPK wildtype (WT) primary hepatocytes, but not in AMPK^{-/-} primary hepatocytes isolated from liver-specific AMPK α 1 and α 2 double knockout (AMPK DKO) mice (revised Fig. 7A), which was generated by crossing the floxed AMPK α 1 and α 2 mice¹ with albumin-Cre recombinase transgenic mice. Taken together, these data demonstrate that AMPK activation is sufficient and required for

induction of Insig in hepatocytes. Moreover, we have removed the description that AMPK increases Insig activity in vivo in the revised manuscript.

3. In Figure 4, Metformin is shown to decrease Insig-gp78 interaction. However, the treatment time is rather long (8h) for a direct effect of AMPK on this interaction. Is this treatment time required for the effect? Metformin is likely to activate AMPK within 1-2h of treatment.

Answer: Thanks for the reviewer's comments. As shown in Fig. 3C, the half-life for Insig-1 protein is around 30 min, which is consistent with the previous observation in SRD-13A cells², whereas proteasomal degradation of Insig-1 is at least 15 times more rapidly than Insig-2². In Fig. 4C and 4D, HEK293 cells were treated with AMPK agonist A769662 for as long as 8 hours to maximize the efficacy of drug treatment on the interaction between Insig and gp78. We agree with the reviewer that the inhibitory effects of A769662 or metformin on the interaction between Insig-1 and gp78 would happen rapidly.

4. Figure 6 evaluates how SREBP processing is affected by AMPK activity. The authors do this by looking at precursor and nuclear SREBP1 that is overexpressed in HepG2 cells. This figure would benefit greatly if the authors instead performed cellular fractionation to look at this. This is especially true in blots where both the precursor and the nuclear SREBP1 go down (in figure 6D, for example), where it is harder to be confident that processing was inhibited (rather than simply the total levels going down). In addition, nuclear fractionation would allow detection of endogenous nuclear SREBP. The authors should do an experiment, such as the one on figure 6A, with cellular fractionation and preferably looking at endogenous SREBP1 and Insig. Additionally, the quantification shown in Fig 6C should analyze the ratio of nSREBP/pSREBP to normalize for variation in expression level.

Answer: We would like thank Reviewer#2 for thoughtful and constructive comments. First, given that the positive feedback regulation of SREBP through the SRE element in its own promoter³. It is technically difficult to determine whether the change of nuclear SREBP fragment is due to the regulation of proteolytic cleavage or transcription. We therefore

Fig. 1. The measurement of both nuclear and full length SREBP-1 fragment in HepG2 hepatocytes stably expressing SREBP-1c. The cell lysates were individually immunoblotted with an antibody against FLAG and three different antibodies against SREBP-1 as indicated.

sought to generate exogenous FLAG-tagged SREBP expression system, in which FLAG-tagged SREBP-1c is driven by a CMV promoter. The specificity of both nuclear and precursor SREBP-1c is confirmed by FLAG antibody and all three different SREBP-1 antibodies (Fig. 1). Notably, the FLAG antibody appears to have the highest sensitivity. Second, the cellular fractionation was performed using the SREBP-1c stable hepatocytes. As shown in revised Fig. 6B, consistent with the measurement using whole lysates, the proteolytic processing and nuclear fragment of exogenous SREBP-1c were strikingly decreased by treatment with metformin (2 mM), whereas the expression of precursor SREBP-1c was not obviously changed. In addition, these results also suggest that the use of HepG2 cells stably expressing FLAG-tagged SREBP-1c is a unique and valuable tool for the study of SREBP-1 proteolytic cleavage and hepatic lipogenesis in vivo.

The mild reduction of precursor SREBP-1c in the submitted manuscript Fig. 6D are likely due to the condition of the cells, or the treatment with adenoviruses expressing GFP or Insig-1. Moreover, due to the limited space, we have moved these data in the supplemental section

(revised Fig. S4). Moreover, the ratio of nuclear to precursor SREBP levels has been quantified and presented as nSREBP-1c/pSREBP-1c in revised Fig. 6D.

5. In figure 6F, the authors claim that the very last picture has less Oil Red O than the previous picture (metformin-treated SREBP-1c-S372A cells have less lipids than vehicle control). This is not obvious at all from the picture. Please quantify the Oil Red O staining, so the result can be visualized better.

Answer: Thanks for the comments. The quantification of Oil Red O staining of HepG2 cells stably expressing empty vector, SREBP-1c or SREBP-1c S372A was performed as described previously^{4, 5} and presented in revised Fig. 6I. Consistent with the cleavage assay in revised Fig. 6C and 6D, SREBP-1c S372A mutant stably expressed in human HepG2 cells remains responsive to metformin-inhibited proteolytic cleavage, further supporting the notion that AMPK phosphorylates Insig and inhibits SREBP activity to attenuate hepatic steatosis.

6. In Figure 7, the authors show that Insig-1 overexpression in AMPKα2 LKO rescues some features of steatosis. However, this experiment is somewhat misleading, as increasing Insig expression is likely to correct liver steatosis in WT as well (which is not shown here). Therefore, the connection to the specific phenotype of AMPKα2 LKO is hard to understand. More should be done to show that Insig1 is necessary for the effect of Metformin and direct AMPK activators on SREBP processing. For instance, CRISPR KO cell lines for AMPK and Insig could be generated in HepG2 and their contribution to the effect of Metformin on SREBP processing as well as on lipogenic gene expression should be evaluated (similarly to Figure 7G). Alternatively, if the authors have access to AMPK-null liver (AMPKα2 LKO likely still have AMPKα1 and residual AMPK activity) they could evaluate the effect of Metformin on Insig stabilization in the absence of AMPK, which would really establish AMPK as required for this phenomenon in vivo.

Answer: We would like to thank Reviewer#2 for the insightful and constructive comments. First, as shown in revised Fig. 7E, hepatic steatosis in AMPKα2 LKO mice fed with HFHS diet was significantly reduced by adenovirus-mediated expression of Insig-1, suggesting that Insig reduction may mediate the steatotic phenotypes in AMPKα2 LKO mice. Due to the technical limitation, we hope that the reviewer would agree with us that these data provide additional in vivo evidence to support the notion that AMPK inhibits hepatic lipogenesis via activation of Insig. Second, as shown in revised Fig. 6E, treatment with Insig-1 is sufficient to inhibit FLAG-tagged SREBP-1c cleavage in HepG2 cells and leads to a reduction of its target lipogenic enzyme FAS expression, whereas the T222A mutant does not. These data suggest that AMPK phosphorylation site at T222 is necessary for Insig to inhibit SREBP-1c cleavage and lipogenic gene expression. Third, we have previously demonstrated the relative contribution of AMPK to metformin's inhibitory effects on SREBP cleavage and lipogenesis⁶. Fourth, the half-life of Insig-1 is increased to 1.4 hours by treatment with AMPK agonist A769662 (revised Fig. 3C). Moreover, the induction was ablated by the nonphosphorylatable T222A mutant, and the mutant is also resistant to A769662-induced stabilization of Insig-1 (revised Fig. 5G). Most importantly, activation of AMPK by metformin, A769662 and AICAR enhances protein levels of Insig-1 in WT hepatocytes, whereas these effects are abrogated in AMPKα1α2 double knockout mouse primary hepatocytes (revised Fig. 7A). These data characterize AMPK as an upstream kinase of Insig-1, and demonstrate that AMPK is sufficient and necessary for metformin on the stabilization and activation of Insig. Taken together, these results suggest that Insig may partially mediate AMPK's effects on the regulation of lipogenesis.

Minor points:

1. On page 4, line 1: the authors say "We have previously identified that AMPK is a direct upstream kinase of SREBP. AMPK phosphorylation of SREBP-1c at ser372 site is sufficient and required for the inhibition of proteolytic cleavage and nuclear translocation of SREBP-1c15. However, SREBP-1c S372A mutation remains responsive to

AMPK-mediated proteolytic cleavage and maturation of SREBP-1c, albeit the extent is less than wild-type (WT) SREBP-1c.”

This sounds contradictory. First sentence says that AMPK inhibits cleavage, but the second sentence says “AMPK-mediated proteolytic cleavage”. Is it not the model that AMPK inhibits cleavage? What do the authors mean by “AMPK-mediated cleavage”? Similar comment for page 4, second paragraph: “AMPK-induced proteolytic cleavage in hepatocytes”. Please clarify.

Answer: Thanks for the comments. We have carefully checked the manuscript and revised the errors accordingly.

2. Page 4, last paragraph. The authors should not refer to their metformin treatment as “acute”. 8 weeks of metformin on diet is best described as “sustained”.

Answer: Thanks for the comments. We have removed “acute” in the revised manuscript.

3. Page 7, last paragraph: “kinase assay”. The authors did not perform a kinase assay, but simply a western blot. Please correct.

Answer: Thanks for the comments. We have revised the manuscript accordingly.

4. page 8, line 1. The authors say “... leads to an augmentation of Insig-1 activity and reduction of fatty acid synthesis”. The authors have not evaluated Insig-1 activity or fatty acid synthesis at this point in the paper. They do later, so a claim like that should appear in the text only after the data on Figures 6 and 7 are discussed.

Answer: Thanks for the comments. The description has been replaced as “These results demonstrate that Insig-1 is regulated by AMPK through phosphorylation.” in the manuscript.

5. Typo: page 4, second paragraph: change “has” to “have”.

Answer: Thanks for the comments. We have corrected the error.

6. Typo: page 5, line 1: change “straining” to “staining”.

Answer: Thanks for the comments. We have corrected the error.

7. Typo: page 5, paragraph 3: change “metformin were” to “metformin was”.

Answer: Thanks for the comments. We have corrected the error.

8. Typo: page 7, end of page: “were evidence”. Change. Do they mean “evident”? Evidence of what?

Answer: Thanks for the comments. The description has been replaced as “were evidenced” in the manuscript.

9. Typo: page 9: change “attenuate” to “attenuates”

Answer: Thanks for the comments. We have corrected the error.

10. Typo: page 11, first paragraph: The sentence “To test the hypothesis that activation of AMPK...” is either incomplete (it doesn't say what will be done to test the hypothesis) or just wrong. Please change.

Answer: Thanks for the comments. The description has been replaced as “the findings that the association between AMPK α subunit and Insig-1 is enhanced by AMPK agonist A769662, suggest that activation of AMPK may enable its substrates to be more accessible to the kinase domain of AMPK α .” in the manuscript.

Reviewer #3: Han and colleagues have investigated a possible new mode of SREBP cleavage regulation via AMPK-mediated phosphorylation of Insig. They first show in a DIO model that chronic administration of metformin results in increased Insig-1 and -2 protein levels. This was associated with reduced liver TG, cholesterol and FAS expression. They then turn to in vitro studies using transfection to show AMPK activation leads to increased Insig-1 and -2 protein levels. Further studies show that ubiquitination of Insig is reduced by AMPK activation as is protein degradation. Overexpression studies also suggested that AMPK activation partially attenuates the known role of gp78 in Insig degradation. Co-IP studies again with overexpressed protein suggest that AMPK and Insig directly interact and an activator of AMPK increased Insig phosphorylation. TO assess functional Insig, cells stably transfected with full length SREBP-1c were treated with metformin, which reduced nuclear SREBP-1 protein levels.

Finally, Insig overexpression in AMPKa2 KO mice reduced liver TGs.

Answer: We would like thank Reviewer#3 for the comments.

Major Points:

1. Fig. 1 needs nuclear SREBP-1 and SREBP-2 measurements.

Answer: Thanks for the comments. As requested, immunoblotting analysis was performed to measure expression levels of nuclear SREBP-1. As shown in revised Fig. 1E and Fig. S1C, AMPK agonist metformin treatment caused a reduction of nuclear SREBP protein levels in the liver of mice fed with HFHS diet, which is consistent with an induction of Insig. These data support the notion that AMPK increases Insig activity and inhibits SREBP to attenuate hepatic lipogenesis.

2. Fig. S1- Although not entirely clear, it appears the 293 cells were transfected with a plasmid encoding Insig and then treated with metformin. The mRNA for Insig did not change with treatment so the conclusion is that metformin does not change the mRNA levels despite the increase in protein levels. This is true in the transfected state but one would predict that endogenous mRNA levels of Insig1 would decrease since nuclear SREBP-1 should be reduced.

Answer: We would like to thank Reviewer#3 for the insightful comments. As shown in revised Fig. S1A, mRNA levels of Insig-1 were not obviously changed by metformin in HEK293 cells, suggesting that AMPK activator metformin may not regulate transcription of Insig-1. In response to reviewer's question regarding the regulation of SREBP on Insig-1, real-time PCR was performed to measure SREBP expression in these cells. New experiments show that SREBP-1c mRNA levels were not significantly affected by metformin although a mild reduction was observed (Fig. 2). Although the data appears to be contradictory to the previous findings showing that Insig-1 is positively regulated by SREBP^{7,8}, it is likely due to the low levels of SREBP expression in HEK293 cells at basal condition. Also it is likely due to the cell-type specific regulation of SREBP on Insig-1.

Fig. 2. Effects of metformin on the mRNA levels of SREBP-1c. HEK293 cells were treated with various doses of metformin for 24h, followed by real-time PCR analysis.

3. The final in vivo study is not proof that lack of Insig is responsible for the phenotype. It is likely that Insig overexpression in WT mice fed the HFHS diet would also reduce liver TGs.

Answer: We would like to thank Reviewer#3 for the insightful questions. Please see the same answer for question #6 raised by Reviewer#2.

4. The studies would be more convincing if the phosphorylation site(s) were identified and mutants studied.

Answer: We would like thank Reviewer#3 for thoughtful and constructive comments. Through integrative mass spectrometry proteomic and genetic analysis, we identified that Thr222 phosphorylation of Insig-1 by AMPK is necessary for augmentation of Insig-1 activity and its

inhibitory effects on SREBP-1c cleavage and lipogenic gene expression. Please see the same answer to the question also raised by Reviewer#1.

Minor Points:

1. In general the abstract is not very clear for the average reader and the first and second sentences in particular are too confusing.

Answer: Thanks for the comments. We have revised the manuscript accordingly.

2. It is repeatedly stated there are clinical studies that indicate metformin may be beneficial for humans with NASH- actually all of the studies suggest there is no significant benefit for NASH and no further trials are ongoing.

Answer: Thanks for the comments. We have removed the description of metformin's effects on NASH from the manuscript.

Reference

1. Wu L, Zhang L, Li B, Jiang H, Duan Y, Xie Z, Shuai L, Li J, Li J. Amp-activated protein kinase (ampk) regulates energy metabolism through modulating thermogenesis in adipose tissue. *Frontiers in Physiology*. 2018;9
2. Lee JN, Gong Y, Zhang X, Ye J. Proteasomal degradation of ubiquitinated insig proteins is determined by serine residues flanking ubiquitinated lysines. *Proceedings of the National Academy of Sciences of the United States of America*. 2006;103:4958-4963
3. Amemiya-Kudo M, Shimano H, Yoshikawa T, Yahagi N, Hastay AH, Okazaki H, Tamura Y, Shionoiri F, Iizuka Y, Ohashi K, Osuga J-i, Harada K, Gotoda T, Sato R, Kimura S, Ishibashi S, Yamada N. Promoter analysis of the mouse sterol regulatory element-binding protein-1c gene. *Journal of Biological Chemistry*. 2000;275:31078-31085
4. Yixuan S, Mingfeng X, Hongmei Y, Yamei H, Feifei Z, Zhimin H, Aoyuan C, Fengguang M, Zhengshuai L, Qi G, Xuqing C, Jing G, Hua B, Yi T, Yu L, Xin G. Berberine attenuates hepatic steatosis and enhances energy expenditure in mice by inducing autophagy and fibroblast growth factor 21. *British Journal of Pharmacology*. 2018;175:374-387
5. Li Y, Wong K, Giles A, Jiang J, Lee JW, Adams AC, Kharitonov A, Yang Q, Gao B, Guarente L, Zang M. Hepatic sirt1 attenuates hepatic steatosis and controls energy balance in mice by inducing fibroblast growth factor 21. *Gastroenterology*. 2014;146:539-549.e537
6. Li Y, Xu S, Mihaylova MM, Zheng B, Hou X, Jiang B, Park O, Luo Z, Lefai E, Shyy John YJ, Gao B, Wierzbicki M, Verbeuren Tony J, Shaw Reuben J, Cohen Richard A, Zang M. Ampk phosphorylates and inhibits srebpl activity to attenuate hepatic steatosis and atherosclerosis in diet-induced insulin-resistant mice. *Cell Metabolism*. 2011;13:376-388
7. Janowski BA. The hypocholesterolemic agent LY295427 up-regulates insig-1, identifying the insig-1 protein as a mediator of cholesterol homeostasis through srebpl. *Proceedings of the National Academy of Sciences*. 2002;99:12675-12680
8. Goldstein JL, DeBose-Boyd RA, Brown MS. Protein sensors for membrane sterols. *Cell*. 2006;124:35-46

REVIEWERS' COMMENTS:

Reviewer #2 (Remarks to the Author):

In this revised manuscript by Li and colleagues, the authors have added considerable new data to address the reviewers concerns, including the mapping of a putative dominant AMPK-dependent phosphorylation site as data in Figure 5 and S3 show.

There is one minor odd aspect of the site identified T222. Prediction software indicates this residue should lie in a transmembrane domain so it is odd this site was detected as phosphorylated. Additionally, the T222 site does not conform well at all the established substrate motif shared in common by all known AMPK substrates (including Srebp1). The S43 and S46 site do conform quite well to the optimal substrate sequence, and by the authors data in Figure 5E, it appears that S46A also blocks metformin-induced increase in Insig1 levels.

It is possible that another AMPK-dependent kinase with a different optimal substrate motif (like ULK1 for example), may be the real kinase for T222. The authors should note these possibilities in the discussion of the paper.

With this text addition to the discussion, the manuscript should be acceptable for publication in Nature Communications.

Reviewer #3 (Remarks to the Author):

The authors have performed additional experiments that significantly strengthen the conclusions of the paper.

REVIEWERS' COMMENTS:**Reviewer #2** (Remarks to the Author):

In this revised manuscript by Li and colleagues, the authors have added considerable new data to address the reviewers concerns, including the mapping of a putative dominant AMPK-dependent phosphorylation site as data in Figure 5 and S3 show.

There is one minor odd aspect of the site identified T222. Prediction software indicates this residue should lie in a transmembrane domain so it is odd this site was detected as phosphorylated. Additionally, the T222 site does not conform well at all the established substrate motif shared in common by all known AMPK substrates (including Srebp1). The S43 and S46 site do conform quite well to the optimal substrate sequence, and by the authors data in Figure 5E, it appears that S46A also blocks metformin-induced increase in Insig1 levels.

It is possible that another AMPK-dependent kinase with a different optimal substrate motif (like ULK1 for example), may be the real kinase for T222. The authors should note these possibilities in the discussion of the paper.

With this text addition to the discussion, the manuscript should be acceptable for publication in Nature Communications.

Answer: We would like to thank Reviewer#2 for the positive and insightful comments. We have included “Interestingly, the mutation of S46A blocks metformin-induced Insig-1 levels in Fig. 5E. It is possible that AMPK regulates Insig-1 at other phosphorylation sites. Moreover, we cannot exclude the possibility that Insig-1 is regulated by AMPK dependent kinase indirectly.” in the discussion section of the revised manuscript.

Reviewer #3 (Remarks to the Author):

The authors have performed additional experiments that significantly strengthen the conclusions of the paper.

Answer: We would like to thank Reviewer#3 for the positive comments.